# GOODHART'S LAW IN REINFORCEMENT LEARNING

**Jacek Karwowski**[1]*   **Oliver Hayman**[1]   **Xingjian Bai**[1]   **Klaus Kiendlhofer**[2]
**Charlie Griffin**[1]   **Joar Skalse**[1,3]

[1] University of Oxford   [2] Independent   [3] Future of Humanity Institute

## ABSTRACT

Implementing a reward function that perfectly captures a complex task in the real world is impractical. As a result, it is often appropriate to think of the reward function as a *proxy* for the true objective rather than as its definition. We study this phenomenon through the lens of *Goodhart's law*, which predicts that increasing optimisation of an imperfect proxy beyond some critical point decreases performance on the true objective. First, we propose a way to *quantify* the magnitude of this effect and *show empirically* that optimising an imperfect proxy reward often leads to the behaviour predicted by Goodhart's law for a wide range of environments and reward functions. We then provide a *geometric explanation* for why Goodhart's law occurs in Markov decision processes. We use these theoretical insights to propose an *optimal early stopping method* that provably avoids the aforementioned pitfall and derive theoretical *regret bounds* for this method. Moreover, we derive a training method that maximises worst-case reward, for the setting where there is uncertainty about the true reward function. Finally, we evaluate our early stopping method experimentally. Our results support a foundation for a theoretically-principled study of reinforcement learning under reward misspecification.

## 1 INTRODUCTION

To solve a problem using Reinforcement Learning (RL), it is necessary first to formalise that problem using a reward function (Sutton & Barto, 2018). However, due to the complexity of many real-world tasks, it is exceedingly difficult to directly specify a reward function that fully captures the task in the intended way. However, misspecified reward functions will often lead to undesirable behaviour (Paulus et al., 2018; Ibarz et al., 2018; Knox et al., 2023; Pan et al., 2021). This makes designing good reward functions a major obstacle to using RL in practice, especially for safety-critical applications.

An increasingly popular solution is to *learn* reward functions from mechanisms such as human or automated feedback (e.g. Christiano et al., 2017; Ng & Russell, 2000). However, this approach comes with its own set of challenges: the right data can be difficult to collect (e.g. Paulus et al., 2018), and it is often challenging to interpret it correctly (e.g. Mindermann & Armstrong, 2018; Skalse & Abate, 2023). Moreover, optimising a policy against a learned reward model effectively constitutes a distributional shift (Gao et al., 2023); i.e., even if a reward function is accurate under the training distribution, it may fail to induce desirable behaviour from the RL agent.

Therefore in practice it is often more appropriate to think of the reward function as a *proxy* for the true objective rather than *being* the true objective. This means that we need a more principled understanding of what happens when a proxy reward is maximised, in order to know how we should expect RL systems to behave, and in order to design better algorithms. For example, we aim to answer questions such as: *When is a proxy safe to maximise without constraint? What is the best way to maximise a misspecified proxy? What types of failure modes should we expect from a misspecified proxy?* Currently, the field of RL largely lacks rigorous answers to these types of questions.

In this paper, we study the effects of proxy misspecification through the lens of *Goodhart's law*, an informal principle often stated as "any observed statistical regularity will tend to collapse once pressure is placed upon it for control purposes" (Goodhart, 1984), or more simply: "when a measure becomes a target, it ceases to be a good measure". For example, a students' knowledge of a subject

---

*Correspondence to `jacek.karwowski@cs.ox.ac.uk`

may be correlated with their ability to pass exams on that subject by default. However, students who have sufficiently strong incentives to do well in exams may also include strategies such as cheating for increasing their test score without increasing their understanding. In the context of RL, we can think of a misspecified proxy reward as a measure correlated, but not robustly aligned, with the true objective across some distribution of policies. Goodhart's law then says, informally, that we should expect optimisation of the proxy to initially lead to improvements on the true objective, up until a point where the correlation between the proxy reward and the true objective breaks down, after which further optimisation should lead to worse performance according to the true objective (Figure 1).

In this paper, we present several novel contributions. First, we show that "Goodharting" occurs with high probability for a wide range of environments and pairs of true and proxy reward functions. Next, we provide a mechanistic explanation of *why* Goodhart's law emerges in RL. We use this to derive two new policy optimisation methods and show that they *provably* avoid Goodharting. Finally, we evaluate these methods empirically. We thus contribute towards building a better understanding of the dynamics of optimising towards imperfect proxy reward functions, and show that these insights may be used to design new algorithms.

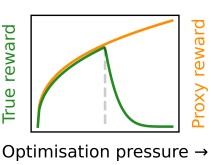

Figure 1: A cartoon of Goodharting.

## 1.1 RELATED WORK

Goodhart's law was first introduced by Goodhart (1984), and has later been elaborated upon by works such as Manheim & Garrabrant (2019). Goodhart's law has also previously been studied in the context of machine learning. In particular, Hennessy & Goodhart (2023) investigate Goodhart's law analytically in the context where a machine learning model is used to evaluate an agent's actions – unlike them, we specifically consider the RL setting. Ashton (2021) shows by example that RL systems can be susceptible to Goodharting in certain situations. In contrast, we show that Goodhart's law is a robust phenomenon across a wide range of environments, explain why it occurs in RL, and use it to devise new solution methods.

In the context of RL, Goodhart's law is closely related to *reward gaming*. Specifically, if reward gaming means an agent finding an unintended way to increase its reward, then Goodharting is an *instance* of reward gaming where optimisation of the proxy initially leads to desirable behaviour, followed by a decrease after some threshold. Krakovna et al. (2020) list illustrative examples of reward hacking, while Pan et al. (2021) manually construct proxy rewards for several environments and then demonstrate that most of them lead to reward hacking. Zhuang & Hadfield-Menell (2020) consider proxy rewards that depend on a strict subset of the features which are relevant to the true reward and then show that optimising such a proxy in some cases may be arbitrarily bad, given certain assumptions. Skalse et al. (2022) introduce a theoretical framework for analysing reward hacking. They then demonstrate that, in any environment and for any true reward function, it is impossible to create a non-trivial proxy reward that is guaranteed to be unhackable. Also relevant, Everitt et al. (2017) study the related problem of reward corruption, Song et al. (2019) investigate overfitting in model-free RL due to faulty implications from correlations in the environment, and Pang et al. (2022) examine reward gaming in language models. Unlike these works, we analyse reward hacking through the lens of *Goodhart's law* and show that this perspective provides novel insights.

Gao et al. (2023) consider the setting where a large language model is optimised against a reward model that has been trained on a "gold standard" reward function, and investigate how the performance of the language model according to the gold standard reward scales in the size of the language model, the amount of training data, and the size of the reward model. They find that the performance of the policy follows a Goodhart curve, where the slope gets less prominent for larger reward models and larger amounts of training data. Unlike them, we do not only focus on language, but rather, aim to establish to what extent Goodhart dynamics occur for a wide range of RL environments. Moreover, we also aim to *explain* Goodhart's law, and use it as a starting point for developing new algorithms.

## 2 PRELIMINARIES

A *Markov Decision Process* (MDP) is a tuple $\langle S, A, \tau, \mu, R, \gamma \rangle$, where $S$ is a set of *states*, $A$ is a set of *actions*, $\tau : S \times A \to \Delta(S)$ is a transition function describing the outcomes of taking actions

at certain states, $\mu \in \Delta(S)$ is the distribution of the initial state, $R \in \mathbb{R}^{|S \times A|}$ gives the reward for taking actions at each state, and $\gamma \in [0,1]$ is a time discount factor. In the remainder of the paper, we consider $A$ and $S$ to be finite. Our work will mostly be concerned with rewardless MDPs, denoted by MDP\R $= \langle S, A, \tau, \mu, \gamma \rangle$, where the true reward $R$ is unknown. A *trajectory* is a sequence $\xi = (s_0, a_0, s_1, a_1, \ldots)$ such that $a_i \in A$, $s_i \in S$ for all $i$. We denote the space of all trajectories by $\Xi$. A *policy* is a function $\pi : S \to \Delta(A)$. We say that the policy $\pi$ is deterministic if for each state $s$ there is some $a \in A$ such that $\pi(s) = \delta_a$. We denote the space of all policies by $\Pi$ and the set of all deterministic policies by $\Pi_0$. Each policy $\pi$ on an MDP\R induces a probability distribution over trajectories $\mathbb{P}(\xi | \pi)$; drawing a trajectory $(s_0, a_0, s_1, a_1, \ldots)$ from a policy $\pi$ means that $s_0$ is drawn from $\mu$, each $a_i$ is drawn from $\pi(s_i)$, and $s_{i+1}$ is drawn from $\tau(s_i, a_i)$ for each $i$.

For a given MDP, the *return* of a trajectory $\xi$ is defined to be $G(\xi) := \sum_{t=0}^{\infty} \gamma^t R(s_t, a_t)$ and the expected return of a policy $\pi$ to be $\mathcal{J}(\pi) = \mathbb{E}_{\xi \sim \pi}[G(\xi)]$. An *optimal policy* is one that maximizes expected return; the set of optimal policies is denoted by $\pi_\star$. There might be more than one optimal policy, but the set $\pi_\star$ always contains at least one deterministic policy (Sutton & Barto, 2018). We define the value-function $V^\pi : S \to \mathbb{R}$ such that $V^\pi[s] = \mathbb{E}_{\xi \sim \pi}[G(\xi) | s_0 = s]$, and define the $Q$-function $Q^\pi : S \times A \to \mathbb{R}$ to be $Q^\pi(s, a) = \mathbb{E}_{\xi \sim \pi}[G(\xi) | s_0 = s, a_0 = a]$. $V^\star, Q^\star$ are the value and $Q$ functions under an optimal policy. Given an MDP\R , each reward $R$ defines a separate $V_R^\pi$, $Q_R^\pi$, and $\mathcal{J}_R(\pi)$. In the remainder of this section, we fix a particular MDP\R $= \langle S, A, \tau, \mu, \gamma \rangle$.

## 2.1 THE CONVEX PERSPECTIVE

In this section, we introduce some theoretical constructs that are needed to express many of our results. We first need to familiarise ourselves with the *occupancy measures* of policies:

**Definition 1** (State-action occupancy measure). We define a function $\eta^- : \Pi \to \mathbb{R}^{|S \times A|}$, assigning, to each $\pi \in \Pi$, a vector of *occupancy measure* describing the discounted frequency that a policy takes each action in each state. Formally,

$$\eta^\pi(s, a) = \sum_{t=0}^{\infty} \gamma^t \mathbb{P}(s_t = s, a_t = a \mid \xi \sim \pi)$$

We can recover $\pi$ from $\eta^\pi$ on all visited states by $\pi(s, a) = (1 - \gamma)\eta^\pi(s, a)/\left(\sum_{a' \in A} \eta^\pi(s, a')\right)$. If $\sum_{a' \in A} \eta^\pi(s, a') = 0$, we can set $\pi(s, a)$ arbitrarily. This means that we often can decide to work with the set of possible occupancy measures, rather than the set of all policies. Moreover:

**Proposition 1.** *The set $\Omega = \{\eta^\pi : \pi \in \Pi\}$ is the convex hull of the finite set of points corresponding to the deterministic policies $\{\eta^\pi : \pi \in \Pi_0\}$. It lies in an affine subspace of dimension $|S|(|A| - 1)$.*

Note that $\mathcal{J}_R(\pi) = \eta^\pi \cdot R$, meaning that each reward $R$ induces a linear function on the convex polytope $\Omega$, which reduces finding the optimal policy to solving a linear programming problem in $\Omega$. Many of our results crucially rely on this insight. We denote the orthogonal projection map from $\mathbb{R}^{|S \times A|}$ to $\text{span}(\Omega)$ by $M_\tau$, which means $\mathcal{J}_R(\pi) = \eta^\pi \cdot M_\tau R$, The proof of Proposition 1, and all other proofs, are given in the appendix.

## 2.2 QUANTIFYING GOODHART'S LAW

Our work is concerned with *quantifying* the Goodhart effect. To do this, we need a way to quantify the *distance between rewards*. We do this using the *projected angle between reward vectors*.

**Definition 2** (Projected angle). Given two reward functions $R_0, R_1$, we define $\arg(R_0, R_1)$ to be the angle between $M_\tau R_0$ and $M_\tau R_1$.

The projected angle distance is an instance of a *STARC metric*, introduced by Skalse et al. (2023a).[1] Such metrics enjoy strong theoretical guarantees and satisfy many desirable desiderata for reward function metrics. For details, see Skalse et al. (2023a). In particular:

**Proposition 2.** *We have $\arg(R_0, R_1) = 0$ if and only if $R_0, R_1$ induce the same ordering of policies, or, in other words, $\mathcal{J}_{R_0}(\pi) \leq \mathcal{J}_{R_0}(\pi') \iff \mathcal{J}_{R_1}(\pi) \leq \mathcal{J}_{R_1}(\pi')$ for all policies $\pi, \pi'$.*

---

[1]In their terminology, the *canonicalisation function* is $M_\tau$, and measuring the angle between the resulting vectors is (bilipschitz) equivalent to normalising and measuring the distance with the $\ell^2$-norm.

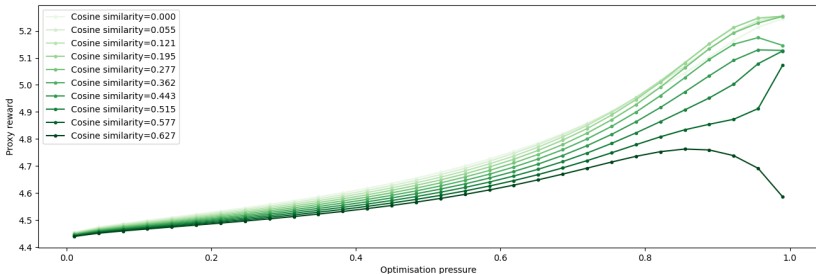

Figure 2: Depiction of Goodharting in *RandomMDP*. Compare to Figure 1 – here we only show the *true* reward obtained by a policy trained on each proxy. Darker color means a more distant proxy.

We also need a way to quantify *optimisation pressure*. We do this using two different training methods. Both are parametrised by *regularisation strength* $\alpha \in (0, \infty)$: Given a reward $R$, they output a regularised policy $\pi_\alpha$. For ease of discussion and plotting, it is often more appropriate to refer to the (bounded) inverse of the regularisation strength: the *optimisation pressure* $\lambda_\alpha = e^{-\alpha}$. As the optimisation pressure increases, $\mathcal{J}(\pi_\alpha)$ also increases.

**Definition 3** (Maximal Causal Entropy). We denote by $\pi_\alpha$ the optimal policy according to the regularised objective $\tilde{R}(s, a) := R(s, a) + \alpha H(\pi(s))$ where $H(\pi(s))$ is the Shannon entropy.

**Definition 4** (Boltzmann Rationality). The Boltzmann rational policy $\pi_\alpha$ is defined as $\mathbb{P}\left(\pi_\alpha(s) = a\right) \propto e^{\frac{1}{\alpha} Q^\star(s,a)}$, where $Q^\star$ is the optimal $Q$-function.

We perform experiments to verify that our key results hold for either way of quantifying optimisation pressure. In both cases, the optimisation algorithm is Value Iteration (see e.g. Sutton & Barto, 2018).

Finally, we need a way to quantify the *magnitude of the Goodhart effect*. Assume that we have a true reward $R_0$ and a proxy reward $R_1$, that $R_1$ is optimised according to one of the methods in Definition 3-4, and that $\pi_\lambda$ is the policy that is obtained at optimisation pressure $\lambda$. Suppose also that $R_0, R_1$ are normalised, so that $\min_\pi \mathcal{J}(\pi) = 0$ and $\max_\pi \mathcal{J}(\pi) = 1$ for both $R_0$ and $R_1$.

**Definition 5** (Normalised drop height). We define the *normalised drop height* (NDH) as $\max_{\lambda \in [0,1]} \mathcal{J}_{R_0}(\pi_\lambda) - \mathcal{J}_{R_0}(\pi_1)$, i.e. as the loss of true reward throughout the optimisation process.

For an illustration of the above definition, see the grey dashed line in Figure 1. We observe that NDH is non-zero if and only if, over increasing optimisation pressure, the proxy and true rewards are initially correlated, and then become anti-correlated (we will see later that as long as the angle distance is less than $\pi/2$, their returns will almost always be initially correlated). In the Appendix C, we introduce more complex measures which quantify Goodhart's law differently. Since our experiments indicate that they are all are strongly correlated, we decided to focus on NDH as the simplest one.

## 3 GOODHARTING IS PERVASIVE IN REINFORCEMENT LEARNING

In this section, we empirically demonstrate that Goodharting occurs pervasively across varied environments by showing that, for a given true reward $R_0$ and a proxy reward $R_1$, beyond a certain optimisation threshold, the performance on $R_0$ *decreases* when the agent is trained towards $R_1$. We test this claim over different kinds of environments (varying number of states, actions, terminal states and $\gamma$), reward functions (varying rewards' types and sparsity) and optimisation pressure definitions.

### 3.1 ENVIRONMENT AND REWARD TYPES

*Gridworld* is a deterministic, grid-based environment, with the state space of size $n \times n$ for parameter $n \in \mathbb{N}^+$, with a fixed set of five actions: $\uparrow, \rightarrow, \downarrow, \leftarrow$, and WAIT. The upper-left and lower-right corners are designated as terminal states. Attempting an illegal action $a$ in state $s$ does not change the state. *Cliff* (Sutton & Barto, 2018, Example 6.6) is a *Gridworld* variant where an agent aims to reach the lower right terminal state, avoiding the cliff formed by the bottom row's cells. Any cliff-adjacent move has a slipping probability $p$ of falling into the cliff.

*RandomMDP* is an environment in which, for a fixed number of states $|S|$, actions $|A|$, and terminal states $k$, the transition matrix $\tau$ is sampled uniformly across all stochastic matrices of shape $|S \times A| \times |S|$, satisfying the property of having exactly $k$ terminal states.

*TreeMDP* is an environment corresponding to nodes of a rooted tree with branching factor $b = |A|$ and depth $d$. The root is the initial state and each action from a non-leaf node results in states corresponding to the node's children. Half of the leaf nodes are terminal states and the other half loop back to the root, which makes it isomorphic to an infinite self-similar tree.

In our experiments, we only use reward functions that depend on the next state $R(s, a) = R(s)$. In **Terminal**, the rewards are sampled iid from $U(0, 1)$ for terminal states and from $U(-1, 0)$ for non-terminal states. In **Cliff**, where the rewards are sampled iid from $U(-5, 0)$ for cliff states, from $U(-1, 0)$ for non-terminal states, and from $U(0, 1)$ for the goal state. In **Path**, where we first sample a random walk $P$ moving only $\rightarrow$ and $\downarrow$ between the upper-left and lower-right terminal state, and then the rewards are constantly 0 on the path $P$, sampled from $U(-1, 0)$ for the non-terminal states, and from $U(0, 1)$ for the terminal state.

### 3.2 Estimating the prevalence of Goodharting

To get an estimate of how prevalent Goodharting is, we run an experiment where we vary all hyperparameters of MDPs in a grid search manner. Specifically, we sample: *Gridworld* for grid lengths $n \in \{2, 3, \ldots, 14\}$ and either **Terminal** or **Path** rewards; *Cliff* with tripping probability $p = 0.5$ and grid lengths $n \in \{2, 3, \ldots, 9\}$ and **Cliff** rewards; *RandomMDP* with number of states $|S| \in \{2, 4, 8, 16, \ldots, 512\}$, number of actions $|A| \in \{2, 3, 4\}$, a fixed number of terminal states $= 2$, and **Terminal** rewards; *TreeMDP* with branching factor 2 and depth $d \in [2, 3, \ldots, 9]$, for two different kinds of trees: (1) where the first half of the leaves are terminal states, and (2) where every second leaf is a terminal state, both using **Terminal** rewards.

For each of those, we also vary temporal discount factor $\gamma \in \{0.5, 0.7, 0.9, 0.99\}$, sparsity factor $\sigma \in \{0.1, 0.3, 0.5, 0.7, 0.9\}$, optimisation pressure $\lambda = -\log(x)$ for 7 values of $x$ evenly spaced on $[0.01, 0.75]$ and 20 values evenly spaced on $[0.8, 0.99]$.

After sampling an MDP\R, we randomly sample a pair of reward functions $R_0$ and $R_1$ from a chosen distribution. These are then sparsified (random $\sigma$ fraction of values are zeroed) and linearly interpolated, creating a sequence of proxy reward functions $R_t = (1 - t)R_0 + tR_1$ for $t \in [0, 1]$. Note that for every environment, reward sampling scheme and fixed choice of parameters considered in Section 3.1, the sample space of rewards is convex. In high dimensions, two random vectors are approximately orthogonal with high probability, so the sequence $R_t$ spans a range of distances.

Each run consists of 10 proxy rewards; we use threshold $\theta = 0.001$ for value iteration. We get a total of 30400 data points. An initial increase, followed by a decline in value with increasing optimisation pressure, indicates Goodharting behaviour. Overall, we find that a Goodhart drop occurs (meaning that the NDH > 0) for **19.3% of all experiments** sampled over the parameter ranges given above. This suggests that Goodharting is a common (albeit not universal) phenomenon in RL and occurs in various environments and for various reward functions. We present additional empirical insights, such as that training *myopic* agents makes Goodharting less severe, in Appendix G.

For illustrative purposes, we present a single run of the above experiment in Figure 2. We can see that, as the proxy $R_1$ is maximised, the true reward $R_0$ will typically either increase monotonically or increase and then decrease. This is in accordance with the predictions of Goodhart's law.

## 4 Explaining Goodhart's Law in Reinforcement Learning

In this section, we provide an intuitive, mechanistic account of *why* Goodharting happens in MDPs, that explains some of the results in Section 3. An extended discussion is also given in Appendix A.

First, recall that $\mathcal{J}_R(\pi) = \eta^\pi \cdot R$, where $\eta^\pi$ is the occupancy measure of $\pi$. Recall also that $\Omega$ is a convex polytope. Therefore, the problem of finding an optimal policy can be viewed as maximising a linear function $R$ within a convex polytope $\Omega$, which is a linear programming problem.

*Steepest ascent* is the process that changes $\vec{\eta}$ in the direction that most rapidly increases $\vec{\eta} \cdot R$ (for a formal definition, see Chang & Murty (1989) or Denel et al. (1981)). The path of steepest ascent

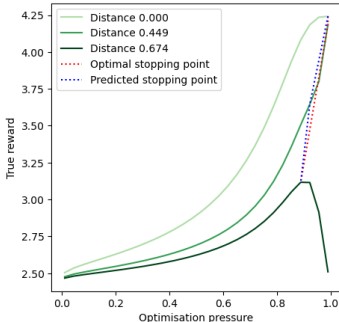
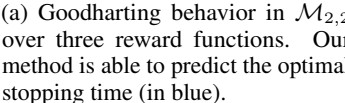

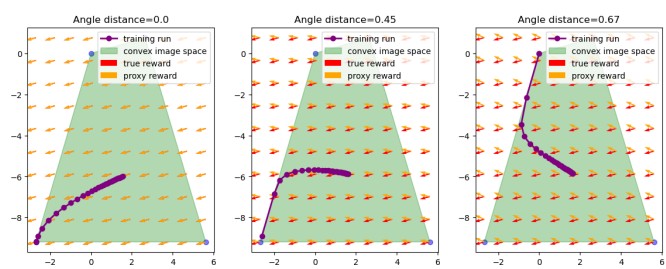

(a) Goodharting behavior in $\mathcal{M}_{2,2}$ over three reward functions. Our method is able to predict the optimal stopping time (in blue).

(b) Training runs for each of the reward functions embedded in the state-action occupancy measure space. Even though the full frequency space is $|S||A| = 4$-dimensional, the image of the policy space occupies only a $|S|(|A| - 1) = 2$-dimensional linear subspace. Goodharting occurs when the cosine distance between rewards passes the critical threshold and the policy snaps to a different endpoint.

Figure 3: Visualisation of Goodhart's law in case of $\mathcal{M}_{2,2}$.

forms a piecewise linear curve whose linear segments lie on the boundary of $\Omega$ (except the first segment, which may lie in the interior). Due to its similarity to gradient-based optimisation methods, we expect most policy optimisation algorithms to follow a path that roughly approximates steepest ascent. Steepest ascent also has the following property:

**Proposition 3** (Concavity of Steepest Ascent). *If $\vec{t}_i := \frac{\eta_{i+1} - \eta_i}{||\eta_{i+1} - \eta_i||}$ for $\eta_i$ produced by steepest ascent on reward vector $R$, then $\vec{t}_i \cdot R$ is decreasing.*

We can now explain Goodhart's law in MDPs. Assume we have a true reward $R_0$ and a proxy reward $R_1$, that we optimise $R_1$ through steepest ascent, and that this produces a sequence of occupancy measures $\{\eta_i\}$. Recall that this sequence forms a piecewise linear path along the boundary of a convex polytope $\Omega$, and that $\mathcal{J}_{R_0}$ and $\mathcal{J}_{R_1}$ correspond to linear functions on $\Omega$ (whose directions of steepest ascent are given by $M_\tau R_0$ and $M_\tau R_1$). First, if the angle between $M_\tau R_0$ and $M_\tau R_1$ is less than $\pi/2$, and the initial policy $\eta_0$ lies in the interior of $\Omega$, then it is guaranteed that $\eta \cdot R_0$ will increase along the first segment of $\{\eta_i\}$. However, when $\{\eta_i\}$ reaches the boundary of $\Omega$, steepest ascent continues in the direction of the projection of $M_\tau R_1$ onto this boundary. If this projection is far enough from $R_0$, optimising in the direction of $M_\tau R_1$ would lead to a decrease in $\mathcal{J}_{R_0}$ (c.f. Figure 3b). *This corresponds to Goodharting.*

$R_0$ may continue to increase, even after another boundary region has been hit. However, each time $\{\eta_i\}$ hits a new boundary, it changes direction, and there is a risk that $\eta \cdot R_0$ will decrease. In general, this is *more* likely if the angle between that boundary and $\{\eta_i\}$ is close to $\pi/2$, and *less* likely if the angle between $M_\tau R_0$ and $M_\tau R_1$ is small. This explains why Goodharting is less likely when the angle between $M_\tau R_0$ and $M_\tau R_1$ is small. Next, note that Proposition 3 implies that the angle between $\{\eta_i\}$ and the boundary of $\Omega$ will increase over time along $\{\eta_i\}$. This explains why Goodharting becomes more likely when more optimisation pressure is applied.

Let us consider an example to make our explanation of Goodhart's law more intuitive. Let $\mathcal{M}_{2,2}$ be an MDP with 2 states and 2 actions, and let $R_0, R_1, R_2$ be three reward functions in $\mathcal{M}_{2,2}$. The full specifications for $\mathcal{M}_{2,2}$ and $R_0, R_1, R_2$ are given in Appendix E. We will refer to $R_0$ as the *true reward*. The angle between $R_0$ and $R_1$ is larger than the angle between $R_0$ and $R_2$. Using Maximal Causal Entropy, we can train a policy over each of the reward functions, using varying degrees of optimisation pressure, and record the performance of the resulting policy with respect to the *true* reward. Zero optimisation pressure results in the uniformly random policy, and maximal optimisation pressure results in the optimal policy for the given proxy (see Figure 3a). As we can see, we get Goodharting for $R_2$ – increasing $R_2$ initially increases $R_0$, but there is a critical point after which further optimisation leads to worse performance under $R_0$.

To understand what is happening, we embed the policies produced during each training run in $\Omega$, together with the projections of $R_0, R_1, R_2$ (see Figure 3b). We can now see that Goodharting must occur precisely when the angle between the true reward and the proxy reward passes the critical

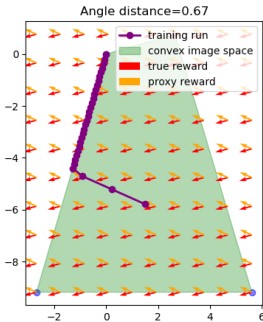

**Angle distance=0.67**

(a) A $\eta$-embedded training run for steepest ascent. The training curve is split into two linear segments: the first is parallel to the proxy reward, while the second is parallel to the proxy reward projected onto some boundary plane $P$. Goodharting only occurs along $P$. (Compare to the MCE approximation of Steepest Ascent in Figure 3b)

**procedure** $\text{EARLYSTOPPING}(S, A, \tau, \theta, R)$
$\quad \vec{r} \leftarrow M_\tau R$
$\quad \pi \leftarrow Unif[\mathbb{R}^{S \times A}]$
$\quad \vec{\eta}_0 \leftarrow \eta^\pi$
$\quad \vec{t}_0 \leftarrow \text{argmax}_{\vec{t} \in T(\vec{\eta}_0)} \vec{t} \cdot R$
$\quad$ **while** $(\vec{t}_i \neq \vec{0})$ **and** $R \cdot \vec{t}_i \leqslant \sin(\theta)\|R\|$ **do**
$\quad\quad \lambda \leftarrow \max\{\lambda : \vec{\eta}_i + \lambda\vec{t}_i \in \Omega\}$
$\quad\quad \vec{\eta}_{i+1} \leftarrow \vec{\eta}_i + \lambda\vec{t}_i$
$\quad\quad \vec{t}_{i+1} \leftarrow \text{argmax}_{\vec{t} \in T(\vec{\eta}_{i+1})} \vec{t} \cdot R$
$\quad\quad i \leftarrow i + 1$
$\quad$ **end while**
$\quad$ **return** $(\eta^{\tilde{\eta}_i})^{-1}$
**end procedure**

(b) Early stopping pseudocode for Steepest Ascent. Given the correct $\theta$, the algorithm would stop at the point where the training run hits the boundary of the convex hull. The cone of tangents, $T(\eta)$ is defined in Denel et al. (1981).

Figure 4: Early stopping algorithm and its behaviour.

threshold, such that the training run deflects upon stumbling on the border of $\Omega$, and the optimal deterministic policy changes from the lower-left to the upper-left corner. *This is the underlying mechanism that produces Goodhart behaviour in reinforcement learning!*

We thus have an explanation for why the Goodhart curves are so common. Moreover, this insight also explains why Goodharting does not always happen and why a smaller distance between the true reward and the proxy reward is associated with less Goodharting. We can also see that Goodharting will be more likely when the angle between $\{\eta_i\}$ and the boundary of $\Omega$ is close to $\pi/2$ – this is why Proposition 3 implies that Goodharting becomes more likely with more optimisation pressure.

## 5 PREVENTING GOODHARTING BEHAVIOUR

We have seen that when a proxy reward $R_1$ is optimised, it is common for the true reward $R_0$ to first increase, and then decrease. If we can stop the optimisation process before $R_0$ starts to decrease, then we can avoid Goodharting. Our next result shows that we can *provably* prevent Goodharting, given that we have a bound $\theta$ on the distance between $R_1$ and $R_0$:

**Theorem 1.** *Let $R_1$ be any reward function, let $\theta \in [0, \pi]$ be any angle, and let $\pi_A, \pi_B$ be any two policies. Then there exists a reward function $R_0$ with $\arg(R_0, R_1) \leqslant \theta$ and $\mathcal{J}_{R_0}(\pi_A) > \mathcal{J}_{R_0}(\pi_B)$ iff*

$$\frac{\mathcal{J}_{R_1}(\pi_B) - \mathcal{J}_{R_1}(\pi_A)}{\|\eta^{\pi_B} - \eta^{\pi_A}\|} < \sin(\theta)\|M_\tau R_1\|$$

**Corollary 1** (Optimal Stopping). *Let $R_1$ be a proxy reward, and let $\{\pi_i\}$ be a sequence of policies produced by an optimisation algorithm. Suppose the optimisation algorithm is concave with respect to the policy, in the sense that $\frac{\mathcal{J}_{R_1}(\pi_{i+1}) - \mathcal{J}_{R_1}(\pi_i)}{\|\eta^{\pi_{i+1}} - \eta^{\pi_i}\|}$ is decreasing. Then, stopping at minimal $i$ with*

$$\frac{\mathcal{J}_{R_1}(\pi_{i+1}) - \mathcal{J}_{R_1}(\pi_i)}{\|\eta^{\pi_{i+1}} - \eta^{\pi_i}\|} < \sin(\theta)\|M_\tau R_1\|$$

*gives the policy $\pi_i \in \{\pi_i\}$ that maximizes $\min_{R_0 \in \mathcal{F}_R^\theta} \mathcal{J}_{R_0}(\pi_i)$, where $\mathcal{F}_R^\theta$ is the set of rewards given by $\{R_0 : \arg(R_0, R_1) \leqslant \theta, \|M_\tau R_0\| = \theta\}$.*

Let us unpack the statement of this result. If we have a proxy reward $R_1$, and we believe that the angle between $R_1$ and the true reward $R_0$ is at most $\theta$, then $\mathcal{F}_R^\theta$ is the set of all possible true reward functions with a given magnitude $m$. Note that no generality is lost by assuming that $R_0$ has magnitude $m$, since we can rescale any reward function without affecting its policy order. Now, if we

optimise $R_1$, and want to provably avoid Goodharting, then we must stop the optimisation process at a point where there is no Goodharting for any reward function in $\mathcal{F}_R^\theta$. Theorem 1 provides us with such a stopping point. Moreover, if the policy optimisation process is concave, then Corollary 1 tells us that this stopping point, in a certain sense, is worst-case optimal. By Proposition 3, we should expect most optimisation algorithms to be approximately concave.

Theorem 1 derives an optimal stopping point among a single optimisation curve. Our next result finds the optimum among *all* policies through maximising a regularised objective function.

**Proposition 4.** *Given a proxy reward $R_1$, let $\mathcal{F}_R^\theta$ be the set of possible true rewards $R$ such that $\arg(R, R_1) \leqslant \theta$ and $R$ is normalized so that $||M_\tau R|| = ||M_\tau R_1||$. Then, a policy $\pi$ maximises $\min_{R \in \mathcal{F}_R^\theta} \mathcal{J}_R(\pi)$ if and only if it maximises $\mathcal{J}_{R_1}(\pi) - \kappa ||\eta^\pi|| \sin(\arg(\eta^\pi, R_1))$, where $\kappa = \tan(\theta) ||M_\tau R_1||$. Moreover, each local maximum of this objective is a global maximum when restricted to $\Omega$, giving that this function can be practically optimised for.*

The above objective can be rewritten as $||\vec{\eta}_\parallel|| - \kappa ||\vec{\eta}_\perp||$ where $\vec{\eta}_\parallel, \vec{\eta}_\perp$ are the components of $\eta^\pi$ parallel and perpendicular to $M_\tau R_1$.

Stopping early clearly loses proxy reward, but it is important to note that it may also lose true reward. Since the algorithm is pessimistic, the optimisation stops before *any* reward in $\mathcal{F}_R^\theta$ decreases. If we continued ascent past this stopping point, exactly one reward function in $\mathcal{F}_R^\theta$ would decrease (almost surely), but most other reward function would increase. If the true reward function is in this latter set, then early stopping loses some true reward. Our next result gives an upper bound on this quantity:

**Proposition 5.** *Let $R_0$ be a true reward and $R_1$ a proxy reward such that $\|R_0\| = \|R_1\| = 1$ and $\arg(R_0, R_1) = \theta$, and assume that the steepest ascent algorithm applied to $R_1$ produces a sequence of policies $\pi_0, \pi_1, \ldots \pi_n$. If $\pi_\star$ is optimal for $R_0$, we have that*

$$\mathcal{J}_{R_0}(\pi_\star) - \mathcal{J}_{R_0}(\pi_n) \leqslant \operatorname{diameter}(\Omega) - \|\eta^{\pi_n} - \eta^{\pi_0}\| \cos(\theta).$$

It would be interesting to develop policy optimisation algorithms that start with an initial estimate $R_1$ of the true reward $R_0$ and then refine $R_1$ over time as the ambiguity in $R_1$ becomes relevant. Theorems 1 and 4 could then be used to check when more information about the true reward is needed. While we mostly leave this for future work, we carry out some initial exploration in Appendix F.

## 5.1 Experimental Evaluation of Early Stopping

We evaluate the early stopping algorithm experimentally. One problem is that Algorithm 4b involves the projection onto $\Omega$, which is infeasible to compute exactly due to the number of deterministic policies being exponential in $|S|$. Instead, we observe that using MCE and BR approximates the steepest ascent trajectory.

Using the exact setup described in Section 3.2, we verify that *the early stopping procedure prevents Goodharting in all cases*, that is, employing the criterion from Corollary 1 always results in NDH = 0. Because early stopping is pessimistic, some reward will usually be lost. We are interested in whether the choice of (1) operationalisation of optimisation pressure, (2) the type of environment or (3) the angle distance $\theta$ impacts the performance of early stopping. A priori, we expected the answer to the first question to be negative and the answer to the third to be positive. Figure 5a shows that, as expected, the choice between MCE and Boltzmann Rationality has little effect on the performance. Unfortunately, and somewhat surprisingly, the early stopping procedure can, in general, lose out on a lot of reward; in our experiments, this is on average between $10\%$ and $44\%$, depending on the size and the type of environment. The relationship between the distance and the lost reward seems to indicate that for small values of $\theta$, the loss of reward is less significant (c.f. Figure 5b).

## 6 Discussion

**Computing $\eta$ in high dimensions:** Our early stopping method requires computing the occupancy measure $\eta$. Occupancy measures can be approximated via rollouts, though this approximation may be expensive and noisy. Another option is to solve for $\eta = \eta^\pi$ via $\vec{\eta} = (I - \Pi T)^{-1} \Pi \vec{\mu}$ where $T$ is the transition matrix, $\mu$ is the initial state distribution, and $\Pi_{s,(s,a)} = \mathbb{P}(\pi(s) = a)$. This solution could be approximated in large environments.

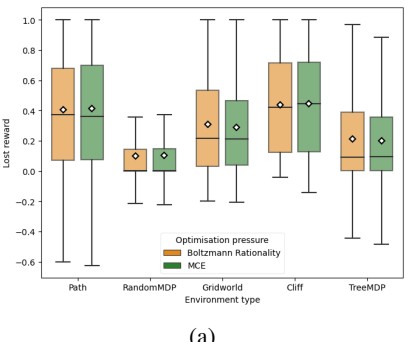
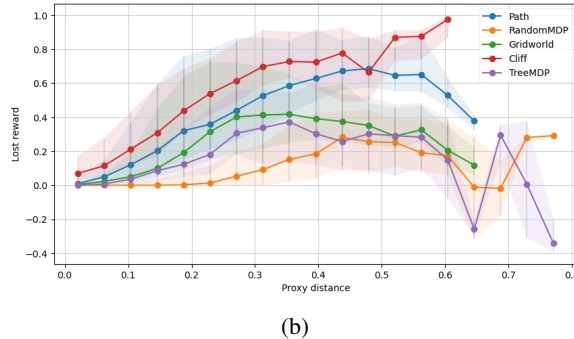

(a)                                                                 (b)

Figure 5: (a) Reward % lost due to the early stopping ($\diamond$ show groups' medians). (b) The relationship between $\theta$ and the lost reward (shaded area between 25th-75th quantiles), aggregated into 25 buckets.

**Approximating $\theta$:** Our early stopping method requires an upper bound $\theta$ on the angle between the true reward and the proxy reward. In practice, this should be seen as a measure of how accurate we believe the proxy to be. If the proxy reward is obtained through reward learning, then we may be able to estimate $\theta$ based on the learning algorithm, the amount of training data, and so on. Moreover, if we have a (potentially expensive) method to evaluate the true reward, such as expert judgement, then we can estimate $\theta$ directly (even in large environments). For details, see Skalse et al. (2023a).

**Key assumptions:** An important consideration when employing any optimisation algorithm is its behaviour when its key assumptions are not met. For our early stopping method, if the provided $\theta$ does not upper-bound the angle between the proxy and the true reward, then the learnt policy may, in the worst case, result in as much Goodharting as a policy produced by naïve optimisation.[2] On the other hand, if the optimisation algorithm is not concave, then this can only cause the early-stopping procedure to stop at a sub-optimal point; Goodharting is still guaranteed to be avoided. This is also true if the upper bound $\theta$ is not tight.

**Significance and Implications:** Our work has several direct implications. In Section 3, we show that Goodharting occurs for a wide range of environments and reward functions. This means that we should expect to see Goodharting often when optimising for misspecified proxy rewards. In Section 4, we provide a mechanistic explanation for *why* Goodharting occurs. We expect this to be helpful for further progress in the study of reward misspecification. In Section 5, we provide early stopping methods that provably avoid Goodharting, and show that these methods, in a certain sense, are worst-case optimal. However, these methods can lead to less true reward than naïve optimisation, This means that they are most applicable when it is essential to avoid Goodharting.

**Limitations and Future Work:** We do not have a comprehensive understanding of the dynamics at play when a misspecified reward function is maximised, and our work does not exhaust this area of study. An important question is what types of failure modes can occur in this setting, and how they may be detected and mitigated. Our work studies one important failure mode (i.e. Goodharting), but there may be other distinctive failure modes that could be described and studied as well. A related important question is precisely how a proxy reward $R_1$ may differ from the true reward $R_0$, before maximising $R_1$ might be bad according to $R_0$. There are several existing results pertaining to this question (Ng et al., 1999; Gleave et al., 2020; Skalse et al., 2022; 2023b), but there is at the moment no comprehensive answer. Another interesting direction is to use our results to develop policy optimisation algorithms that collect more data about the reward function over time, as this information is needed. We discuss this direction in Appendix F. Finally, it would be interesting to try to find principled relaxations of the methods in Section 5, that attain better practical performance while retaining desirable theoretical guarantees.

---

[2]However, it might still be possible to bound the worst-case performance further using the norm of the transition matrix (defining the geometry of the polytope $\Omega$). This will be an interesting topic for future work.

## ACKNOWLEDGEMENTS

The authors would like to thank Oliver Sourbut, Bogdan Ionut Cirstea, Sam Staton and anonymous reviewers for their detailed feedback on the draft of this paper. This research was conducted and funded as a part of Oxford AI Safety Hub Labs. The first author was supported by a separate grant from Open Philanthropy.

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

## A    A More Detailed Explanation of Goodhart's Law

In this section, we provide an intuitive explanation of why Goodharting occurs in MDPs, that will be more detailed an clear than the explanation provided in Section 4.

First of all, as in Section 4, recall that $\mathcal{J}_R(\pi) = \eta^\pi \cdot R$, where $\eta^\pi$ is the occupancy measure of $\pi$. This means that we can decompose $\mathcal{J}_R$ into two steps, the first of which is independent of $R$, and maps $\Pi$ to $\Omega$, and the second of which is a linear function. Recall also that $\Omega$ is a convex polytope. Therefore, the problem of finding an optimal policy can be viewed as maximising a linear function within a convex polytope $\Omega$. If $R_1$ is the reward function we are optimising, then we can visualise this as follows:

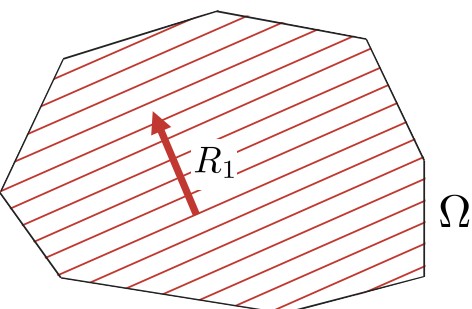

Here the red arrow denotes the direction of $R_1$ within $\Omega$. Note that this direction corresponds to $M_\tau R_1$, rather than $R_1$, since $\Omega$ lies in a lower-dimensional affine subspace. Similarly, the red lines correspond to the level sets of $R_1$, i.e. the directions we can move in without changing $R_1$.

Now, if $R_1$ is a *proxy reward*, then we may assume that there is also some (unknown) true reward function $R_0$. This reward also induces a linear function on $\Omega$:

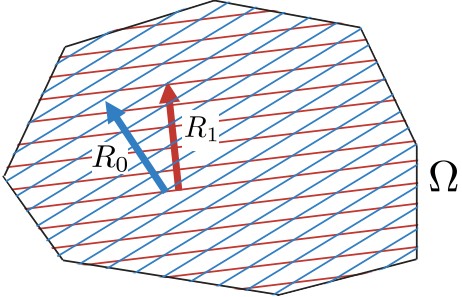

Suppose we pick a random point $\eta^\pi$ in $\Omega$, and then move in a direction that increases $\eta^\pi \cdot R_1$. This corresponds to picking a random policy $\pi$, and then modifying it in a direction that increases $\mathcal{J}_{R_1}(\pi)$. In particular, let us consider what happens to the true reward function $R_0$, as we move in the direction that most rapidly increases the proxy reward $R_1$.

To start with, if we are in the interior of $\Omega$ (i.e., not close to any constraints), then the direction that most rapidly increases $R_1$ is to move parallel to $M_\tau R_1$. Moreover, if the angle $\theta$ between $M_\tau R_1$ and $M_\tau R_0$ is no more than $\pi/2$, then this is guaranteed to also increase the value of $R_0$. To see this, simply consider the following diagram:

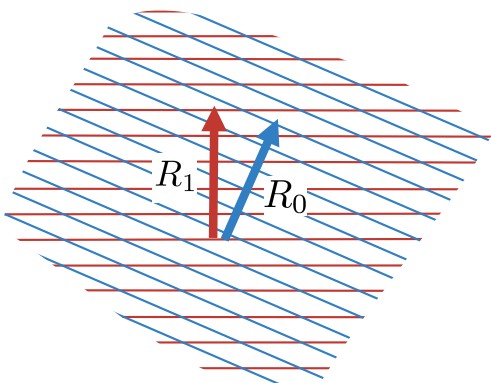

However, as we move parallel to $M_\tau R_1$, we will eventually hit the boundary of $\Omega$. When we do this, the direction that most rapidly increases $R_1$ will no longer be parallel to $M_\tau R_1$. Instead, it will be parallel to the projection of $R_1$ onto the boundary of $\Omega$ that we just hit. Moreover, if we keep moving in *this* direction, then we might no longer be increasing the true reward $R_0$. To see this, consider the following diagram:

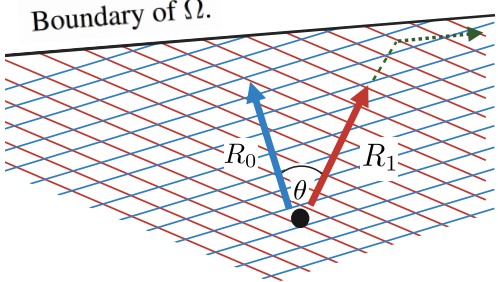

The dashed green line corresponds to the path that most rapidly increases $R_1$. As we move along this path, $R_0$ initially increases. However, after the path hits the boundary of $\Omega$ and changes direction, $R_0$ will instead start to decrease. Thus, if we were to plot $\mathcal{J}_{R_1}(\pi)$ and $\mathcal{J}_{R_0}(\pi)$ over time, we would get a plot that looks roughly like this:

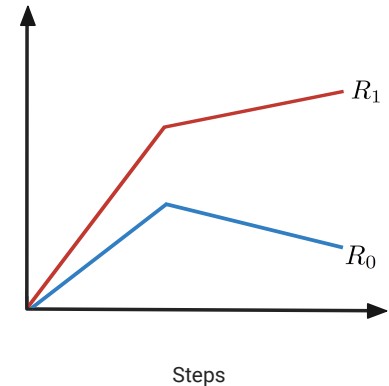

Next, it is important to note that $R_0$ is not guaranteed to decrease after we hit the boundary of $\Omega$. To see this, consider the following diagram:

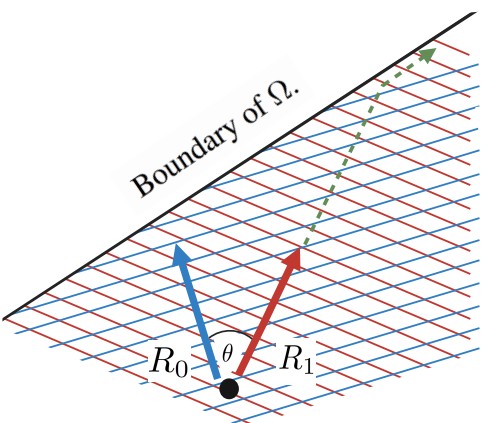

The dashed green line again corresponds to the path that most rapidly increases $R_1$. As we move along this path, $R_0$ will increase both before and after the path has hit the boundary of $\Omega$. If we were to plot $\mathcal{J}_{R_1}(\pi)$ and $\mathcal{J}_{R_0}(\pi)$ over time, we would get a plot that looks roughly like this:

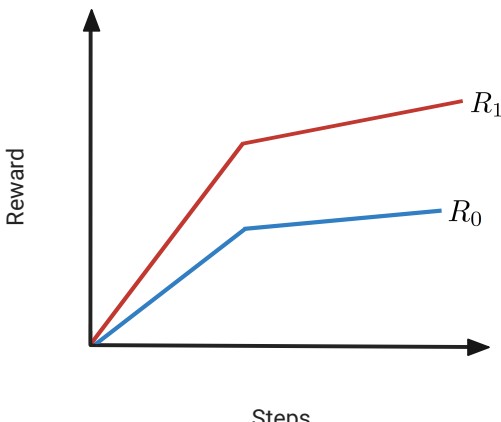

The next thing to note is that we will not just hit the boundary of $\Omega$ once. If we pick a random point $\eta^\pi$ in $\Omega$, and keep moving in the direction that most rapidly increases $\eta^\pi \cdot R_1$ until we have found the maximal value of $R_1$ in $\Omega$, then we will hit the boundary of $\Omega$ over and over again. Each time we hit this boundary we will change the direction that we are moving in, and each time this happens, there is a risk that we will start moving in a direction that decreases $R_0$.

Note that Goodharting corresponds to the case where we follow a path through $\Omega$ along which $R_0$ initially increases, but eventually starts to decrease. As we have seen, this must be caused by the boundaries of $\Omega$. We may now ask; under what conditions do these boundaries force the path of steepest ascent (of $R_1$) to move in a direction that decreases $R_0$? By inspecting the above diagrams, we can see that this depends on the angle between the normal vector of that boundary and $M_\tau R_1$, and the angle between $M_\tau R_1$ and $M_\tau R_0$. In particular, in order for $R_0$ to start decreasing, it has to be the case that the angle between $M_\tau R_1$ and $M_\tau R_0$ is *larger* than the angle between $M_\tau R_1$ and the normal vector of the boundary of $\Omega$. This immediately tells us that if the angle between $M_\tau R_1$ and $M_\tau R_0$ is small (i.e., if $\arg(R_0, R_1)$ is small), then Goodharting will be less likely to occur.

Moreover, as the angle between $M_\tau R_1$ and the normal vector of the boundary of $\Omega$ becomes *smaller*, Goodharting should be correspondingly *more* likely to occur. Next, recall that Proposition 3 tells us that this angle will decrease monotonically along the path of steepest ascent (of $R_1$). As such, Goodharting will get more and more likely, the further we move along the path of steepest ascent. This explains why Goodharting becomes more likely when more optimisation pressure is applied.

## B PROOFS

**Proposition 1.** *The set $\Omega = \{\eta^\pi : \pi \in \Pi\}$ is the convex hull of the finite set of points corresponding to the deterministic policies $\Omega_d := \{\eta^\pi : \pi \in \Pi_0\}$. It lies in a linear subspace of dimension $|S|(|A|-1)$.*

*Proof.* Proof of the second half of this proposition, which says that the dimension of the affine space containing $\Omega$ has at most $|S|(|A|-1)$ dimensions, can be found in (Skalse & Abate, 2023, Lemma A.2). Here, we will prove the first half (that $\Omega = \Omega_d$) using the linear program outlined in Puterman (1994, Equation 6.9.2).

maximise: $R \cdot \eta$

subject to: $\displaystyle\sum_{a \in A} \eta(s', a) - \gamma \sum_{s \in S, a \in A} \tau(s, a, s') \cdot \eta(s, a) = \mu(s)$ $\qquad \forall s' \in S$

$\eta(s, a) \geqslant 0$ $\qquad\qquad\qquad\qquad \forall s, a \in S \times A$

Puterman (1994, Theorem 6.9.1) proves that (i) for any $\pi \in \Pi$, $\eta^\pi$ satisfies this linear program and (ii) for any feasible solution to this linear program $\eta$, there is a policy $\pi$ such that $\eta = \eta^\pi$. In other words, $\Omega = \{\eta \mid \eta \in \mathbb{R}^{|S||A|} A\eta = \mu, \eta \geqslant 0, \}$ where $A$ is an $|S|$ by $|S||A|$ matrix.

Denote the convex hull of a finite set $X$ as $conv(X)$. We first show that $\Omega = conv(\Omega_d)$. The fact that $conv(\Omega_d) \subseteq \Omega$ follows straight from the fact that $\Omega_d \subseteq \Omega$, and from the fact that $\Omega$ must be convex since it is the set of solutions a set of linear equations.

We show that $\Omega \subseteq conv(\Omega_d)$ by strong induction on

$$k(\eta) := \sum_{s \in S} \max(0, |\{a \in A : \eta(s, a) \geqslant 0\}| - 1)$$

Intuitively, $k(\eta) = 0$ if and only if there is a deterministic policy corresponding to $\eta$ and $k(\eta)$ increases with the number of potential actions available in visited states. The base case of the induction is simple, if $k(\eta) = 0$, then there is a deterministic policy $\pi_d$ such that $\eta = \eta^{\pi_d}$ and therefore $\eta \in \Omega_d \subseteq conv(\Omega_d)$.

For the inductive step, suppose $\eta' \in conv(\Omega_d)$ for all $\eta' \in \Omega$ with $k(\eta)' < K$ and consider any $\eta$ with $k(\eta) = K$. We will use the following lemma, which is closely related to (Feinberg & Rothblum, 2012, Lemma 6.3).

**Lemma 1.** *For any occupancy measure $\eta$ with $k(\eta) > 0$, let occupancy measure $x$ be a deterministic reduction of $\eta$ if and only if $k(x) = 0$ and, for all $s, a$, if $x(s, a) > 0$ then $\eta(s, a) > 0$. If $x$ is a deterministic reduction of $\alpha$, then there exists some $\alpha \in (0, 1)$ and $y \in \Omega$ such that $\eta = \alpha x + (1-\alpha)y$ and $k(y) < k(\eta)$.*

Intuitively, since a deterministic reduction always exists, lemma 1 says that any occupancy measure corresponding to a stochastic policy can be split into an occupancy measure corresponding to a deterministic policy, and an occupancy measure with a smaller $k$ number. Proof of lemma 1 is easy, choose $\alpha$ to be the maximum value such that $(\eta - \alpha x)(s, a) \geqslant 0$ for all $s$ and $a$, then set $y = \frac{1}{1-\alpha}(\eta - \alpha x)$. For at least one $s, a$, we will have $(\eta - \alpha x)(s, a) = 0$ and therefore $k(y) < k(\eta)$. It remains to show that $y \in \Omega$, but this follows straightforwardly from (Puterman, 1994, Theorem 6.9.1) and the fact that $y \geqslant 0$, and $Ay = \frac{1}{1-\alpha}A(\eta - \alpha x) = \frac{1}{1-\alpha}(b - \alpha b) = b$.

If $k(\eta)' < K$, then by lemma 1, $\eta = \alpha x + (1-\alpha)y$ with $k(x) = 0$ and $k(y) < K$. By inductive hypothesis, since $k(y) < K$, $y \in conv(\Omega_d)$ and therefore $y$ is a convex combination of vectors in $\Omega_d$. Since $k(x) = 0$, we know that $x \in \Omega_d$ and therefore $\alpha x + (1-\alpha)y$ is also a convex combination of vectors in $\Omega_d$. This suffices to show $\eta \in conv(\Omega_d)$.

By induction $\eta \in conv(\Omega_d)$, for all values of $k(\eta)$, and therefore $\Omega \subseteq conv(\Omega_d)$.

$\square$

**Proposition 2.** *We have $\arg(R_0, R_1) = 0$ if and only if $R_0, R_1$ induce the same ordering of policies, or, in other words, $\mathcal{J}_{R_0}(\pi) \leqslant \mathcal{J}_{R_0}(\pi') \iff \mathcal{J}_{R_1}(\pi) \leqslant \mathcal{J}_{R_1}(\pi')$ for all policies $\pi, \pi'$.*

*Proof.* We show that $\arg(R_0, R_1)$ satisfies the conditions of (Skalse & Abate, 2023, Theorem 2.6). Recall that $\arg(R_0, R_1)$ is the angle between $M_\tau R_0$ and $M_\tau R_1$, where $M_\tau$ projects vectors onto $\Omega$. Now, note that two reward functions $R_0, R_1$ induce different policy orderings if and only if the corresponding policy evaluation functions $\mathcal{J}_0, \mathcal{J}_1$ induce different policy orderings. Moreover, recall that for each $i$ $\mathcal{J}_i$ can be viewed as the linear function $R_i \cdot \eta^\pi$ for $\eta^\pi \in \Omega$. Two linear functions $\ell_0, \ell_1$ defined over a domain $D$ which contains an open set induce different orderings if and only if $\ell_0$ and $\ell_1$ have a non-zero angle after being projected onto $D$. Finally, $\Omega$ does contain a set that is open in the smallest affine space which contains $\Omega$, as per Proposition 1. This means that $R_0$ and $R_1$ induce the same ordering of policies if and only if the angle between $M_\tau R_0$ and $M_\tau R_1$ is 0 (meaning that $\arg(R_0, R_1) = 0$). This completes the proof. $\square$

**Proposition 3** (Concavity of Steepest Ascent). *If $\vec{t_i} := \frac{\eta^{\pi_{i+1}} - \eta^{\pi_i}}{||\eta^{\pi_{i+1}} - \eta^{\pi_i}||}$ for $\eta^{\pi_i}$ produced by steepest ascent on reward vector R, $\vec{t_i} \cdot R$ is nonincreasing.*

*Proof.* By the definition of steepest ascent given in Denel et al. (1981), $\vec{t_i}$ will be the unit vector in the "cone of tangents"

$$T(\eta^{\pi_i}) := \{\vec{t} : ||\vec{t}|| = 1, \exists \lambda > 0, \eta^{\pi_i} + \lambda \vec{t} \in \Omega\}$$

that maximizes $\vec{t_i} \cdot R$. This is what it formally means to go in the direction that leads to the fastest increase in reward.

For sake of contradiction, assume $\vec{t_{i+1}} \cdot R > \vec{t_i} \cdot R$, and let $\vec{t_i'} = \frac{\eta^{\pi_{i+2}} - \eta^{\pi_i}}{||\eta^{\pi_{i+2}} - \eta^{\pi_i}||}$. Then

$$\vec{t_i'} \cdot R = \left( \frac{\vec{t_{i+1}} ||\eta^{\pi_{i+2}} - \eta^{\pi_{i+1}}|| + \vec{t_i} ||\eta^{\pi_{i+1}} - \eta^{\pi_i}||}{||\eta^{\pi_{i+2}} - \eta^{\pi_i}||} \right) \cdot R$$

$$\geqslant \left( \frac{\vec{t_{i+1}} ||\eta^{\pi_{i+2}} - \eta^{\pi_{i+1}}|| + \vec{t_i} ||\eta^{\pi_{i+1}} - \eta^{\pi_i}||}{||\eta^{\pi_{i+2}} - \eta^{\pi_{i+1}}|| + ||\eta^{\pi_{i+1}} - \eta^{\pi_i}||} \right) \cdot R > \vec{t_i} \cdot R$$

where the former inequality follows from triangle inequality and the latter follows as the expression is a weighted average of $\vec{t_{i+1}} \cdot R$ and $\vec{t_i} \cdot R$. We also have for $\lambda = ||\eta^{\pi_{i+2}} - \eta^{\pi_i}||$, $\eta^{\pi_i} + \lambda \vec{t_i'} = \eta^{\pi_{i+2}} \in \Omega$. But then $\vec{t_i'} \in T(\eta^{\pi_i})$, contradicting that $\vec{t_i} = \operatorname{argmax}_{T(\eta^{\pi_i})} \vec{t} \cdot R$. $\square$

**Theorem 1** (Optimal Stopping). *Let $R_1$ be any reward function, let $\theta \in [0, \pi]$ be any angle, and let $\pi_A, \pi_B$ be any two policies. Then there exists a reward function $R_0$ with $\arg(R_0, R_1) \leqslant \theta$ and $\mathcal{J}_{R_0}(\pi_A) > \mathcal{J}_{R_0}(\pi_B)$ if and only if*

$$\frac{\mathcal{J}_{R_1}(\pi_B) - \mathcal{J}_{R_1}(\pi_A)}{||\eta^{\pi_B} - \eta^{\pi_A}||} < \sin(\theta) ||M_\tau R_1||$$

*Proof.* Let $\vec{d} := \eta^{\pi_B} - \eta^{\pi_A}$ denote the difference in occupancy measures. The inequality can be rewritten as

$$\exists R \text{ such that } \arg(R, R_1) \leqslant \theta \text{ and } \vec{d} \cdot R < 0 \iff \cos\left(\arg\left(R_1, \vec{d}\right)\right) < \sin(\theta)$$

To show one direction, if $\vec{d} \cdot R < 0$ we have $\vec{d} \cdot M_\tau R < 0$ as $\vec{d}$ is parallel to $\Omega$. This gives $\arg\left(R, \vec{d}\right) > \frac{\pi}{2}$ and

$$\arg\left(R, \vec{d}\right) \leqslant \arg\left(R_1, \vec{d}\right) + \arg(R_0, R_1) \leqslant \arg\left(R_1, \vec{d}\right) + \theta.$$

It follows that $\arg\left(R_1, \vec{d}\right) > \frac{\pi}{2} - \theta$, and thus $\cos\left(\arg\left(R_1, \vec{d}\right)\right) < \sin(\theta)$.

If instead $\cos\left(\arg\left(R_1, \vec{d}\right)\right) < \sin(\theta)$, we have $\arg\left(R, \vec{d}\right) > \frac{\pi}{2} - \theta$. To choose $R$, there will be two vectors $R \in \mathcal{F}_R^\theta$ that lie at the intersection of the plane $\operatorname{span}(\eta^\pi, M_\tau R_1)$ with the cone $\arg(R, R_1) = \theta$. One will satisfy $\arg(R, \eta^\pi) = \arg(R, R_1) + \arg(R_1, \eta^\pi)$ (informally, when $R_1$ lies between $\eta^\pi$ and $R$). Then this $R$ gives

$$\arg\left(R, \vec{d}\right) = \arg(R, R_1) + \arg\left(R, \vec{d}\right) > \theta + \frac{\pi}{2} - \theta = \frac{\pi}{2}$$

so $R \cdot \vec{d} < 0$. $\square$

**Proposition 4.** *Given a proxy reward $R_1$, let $\mathcal{F}_R^\theta$ be the set of possible true rewards $R$ such that $\arg(R, R_1) \leqslant \theta$ and $R$ is normalized so that $||M_\tau R|| = ||M_\tau R_1||$.*

*Then we have that a policy $\pi$ maximises $\min_{R \in \mathcal{F}_R^\theta} \mathcal{J}_R(\pi)$ if and only if it maximises $\mathcal{J}_{R_1}(\pi) - \kappa||\eta^\pi|| \sin(\arg(\eta^\pi, R_1))$, where $\kappa = \tan(\theta)||M_\tau R_1||$.*

*Moreover, each local maximum of this objective is a global maximum when restricted to $\Omega$, giving that this function can be practically optimised for.*

*Proof.* Note that

$$\min_{R \in \mathcal{F}_R^\theta} \mathcal{J}_R(\pi) = \eta^\pi \cdot M_\tau R = ||M_\tau R_1||||\eta^\pi|| \left( \min_{R \in \mathcal{F}_R^\theta} \cos(\arg(\eta^\pi, R)) \right)$$

as $||M_\tau R_1|| = ||M_\tau R_0||$ for all $R_0$. Now we claim

$$\min_{R \in \mathcal{F}_R^\theta} \cos(\arg(R, \eta^\pi)) = \cos(\arg(R_1, \eta^\pi) + \theta).$$

To show this, we can take $R \in \mathcal{F}_R^\theta$ with $\arg(R, \eta^\pi) = \arg(R_1, \eta^\pi) + \theta$ (such an $R$ is described in appendix B). This then gives

$$\min_{R \in \mathcal{F}_R^\theta} \cos(\arg(R, \eta^\pi)) \leqslant \cos(\arg(R, \eta^\pi) + \theta).$$

We also have

$$\cos(\arg(R, \eta^\pi)) \geqslant \cos(\arg(R_1, \eta^\pi) + \arg(R_1, R)) \geqslant \cos(\arg(R_1, \eta^\pi) + \theta)$$

for any $R$. Then

$$\min_{R \in \mathcal{F}_R^\theta} \cos(\arg(R, \eta^\pi)) = \cos(\arg(R_1, \eta^\pi) + \theta) =$$

$$\cos(\theta) \cos(\arg(R_1, \eta^\pi)) - \sin(\theta) \sin(\arg(R_1, \eta^\pi)).$$

Rearranging gives

$$\min_{R \in \mathcal{F}_R^\theta} \mathcal{J}_R(\pi) \propto R_1 \cdot \eta^\pi - \tan\theta ||\eta^\pi||||M_\tau R_0|| \sin(\arg(R_1, \eta^\pi))$$

which is equivalent to the given objective.

To show that all local maxima are global maxima, note that $\min_{R \in \mathcal{F}_R^\theta} \mathcal{J}_R(\pi) = \min_{R \in \mathcal{F}_R^\theta} \eta^\pi \cdot R$ in $\Omega$ is a minimum over linear functions, and is therefore convex. This then gives that each local maximum of $\min_{R \in \mathcal{F}_R^\theta} \mathcal{J}_R(\pi)$ is a global maximum, so the same holds for the given objective function. □

**Proposition 5.** *Let $R_0$ be a true reward and $R_1$ a proxy reward such that $\|R_0\| = \|R_1\| = 1$ and $\arg(R_0, R_1) = \theta$, and assume that the steepest ascent algorithm applied to $R_1$ produces a sequence of policies $\pi_0, \pi_1, \ldots \pi_n$. If $\pi_\star$ is optimal for $R_0$, we have that*

$$|\mathcal{J}_{R_0}(\pi_n) - \mathcal{J}_{R_0}(\pi_\star)| \leqslant \operatorname{diameter}(\Omega) - \|\eta^{\pi_n} - \eta^{\pi_0}\| \cos(\theta).$$

*Proof.* The bound is composed of two terms: (1) how much total reward $R$ is there to gain, and (2) how much did we gain already. Since the reward vector is normalised, the range of reward over the $\Omega$ is its diameter. The gains that had already been made by the Steepest Ascent algorithm equal $\|\eta^{\pi_n} - \eta^{\pi_0}\|$, but this has to be scaled by the (pessimistic) factor of $\cos(\theta)$, since this is the alignment of the true and proxy reward. □

The bound can be difficult to compute exactly. A simple but crude approximation of the diameter is

$$\max_{\eta_1, \eta_1 \in \Omega} \|\eta_1 - \eta_2\|_2 \leqslant 2 \max_{\eta \in \Omega} \|\eta\|_2 \leqslant \frac{2}{1 - \gamma}$$

## C    MEASURING GOODHARTING

While Goodhart's law is qualitatively well-described, a quantitative measure is needed to systematically analyse it. We propose a number of different metrics to do that. Below, implicitly assuming that all rewards are normalised (as in the Section 2), we denote by $f : [0,1] \rightarrow [0,1]$ the true reward obtained by a policy trained on a *proxy reward*, as a function of optimisation pressure $\lambda$, similarly by $f_0$ the true reward obtained by a policy trained on the *true reward*, and $\lambda^\star = \arg\max_{\lambda \in [0,1]} f(\lambda)$.

- **Normalised drop height:**
$$\mathrm{NDH}(f) = f(1) - f(\lambda^\star)$$

- **Simple integration:**
$$\mathrm{SI}(f) = \left( \int_0^{\lambda^\star} f(\lambda) d\lambda \right) \left( \int_{\lambda^\star}^1 f(\lambda) d\lambda \right)$$

- **Weighted correlation-anticorrelation:**
$$\mathrm{CACW}(f) = -\max(\rho_0, 0)\max(\rho_1, 0)\sqrt{\lambda^\star(1-\lambda^\star)}$$
    where $\rho^i = \rho(f(I_i), I_i)$ are the Pearson correlation coefficients for $I_0 \sim Unif[0, \lambda^\star]$, $I_1 \sim Unif[\lambda^\star, 1]$.

- **Regression angle:**
$$\mathrm{LR}(f) = -\beta_0^+ \beta_1^+$$
    where $\beta_0, \beta_1$ are the angles of the linear regression of $f$ on $[0, \lambda^\star]$ and $[\lambda^\star, 1]$ respectively.

- **Relative weighted integration:**
$$\mathrm{RWI}(f) = \left( (1-\lambda^\star) \int_0^{\lambda^\star} |f(\lambda) - f_0(\lambda)| d\lambda \right) \left( \frac{1}{1-\lambda^\star} \int_{\lambda^\star}^1 |f(\lambda) - f_0(\lambda)| d\lambda \right)$$

The metrics were independently designed to capture the intuition of a *sudden drop in reward with an increased optimisation pressure*. We then generated a dataset of 40000 varied environments:

- *Gridworld*, **Terminal** reward, with $|S| \sim Poiss(100)$, N=1000
- *Cliff*, **Cliff** reward, with $|S| \sim Poiss(100)$, N=500
- *RandomMDP*, **Terminal** reward, $|S| \sim Poiss(100), |A| \sim Poiss(6)$, N=500
- *RandomMDP*, **Terminal** reward, $|S| \sim Unif(16, 64), |A| \sim Unif(2, 16)$, N=500
- *RandomMDP*, **Uniform** reward, $|S| \sim Unif(16, 64), |A| \sim Unif(2, 16)$, N=500
- *CyclicMDP*, **Terminal** reward, depth $\sim Poiss(3)$, N=1000

We have manually verified that all metrics seem to activate strongly on graphs that we would intuitively assign a high degree of Goodharting. In Figure 7, we show, for each metric, the top three training curves from the dataset that each metric assigns the highest score.

We find that all of the metrics are highly correlated - see Figure 6. Because of this, we believe that it is meaningful to talk about a quantitative Goodharting score. Since normalised drop height is the simplest metric, we use it as the proxy for Goodharting in the rest of the paper.

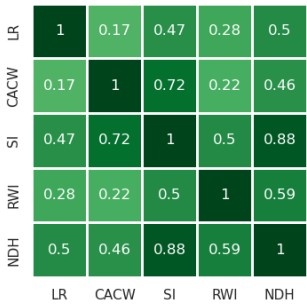

Figure 6: Correlations between different Goodharting metrics, computed over examples where the drop occurs for $\lambda > 0.3$, to avoid selecting adversarial examples.

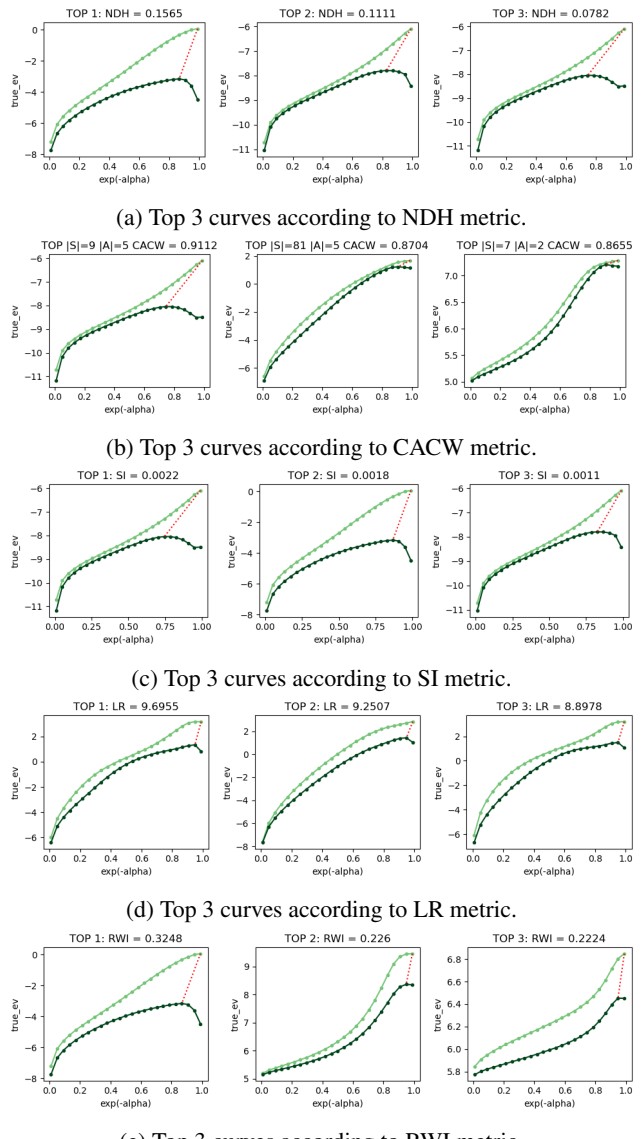

(a) Top 3 curves according to NDH metric.

(b) Top 3 curves according to CACW metric.

(c) Top 3 curves according to SI metric.

(d) Top 3 curves according to LR metric.

(e) Top 3 curves according to RWI metric.

Figure 7: Examples of training curves that obtain high Goodharting scores according to each metric.

# D    EXPERIMENTAL EVALUATION OF THE EARLY STOPPING ALGORITHM

To sanity-check the experiments, we present an additional graph of the relationship between NDH and early stopping algorithm reward loss in Figure 8, and the full numerical data for Figure 5a. We also show example runs of the experiment in all environments in Figure 9.

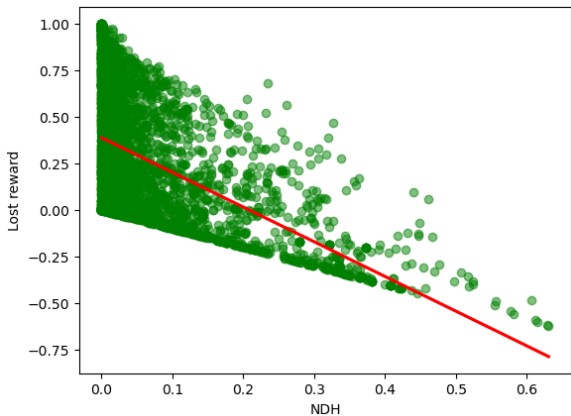

Figure 8: The relationship between the amount of Goodharting, measured by NDH, and the amount of reward that is lost due to pessimistic stopping. As Goodharting increases (measured by an increase in NDH), the potential for *gaining* reward by early stopping increases (fitted linear regression shown as the red line. $y = -1.863x + 0.388$, 95% CI for slope: $[-1.951, -1.775]$, 95% CI for intercept: $[0.380, 0.396]$, $R^2 = 0.23$, $p < 1e - 10$). Only the points with NDH > 0 are shown in the plot.

| Environment | | count | mean | std | min | 25% | 50% | 75% | max |
|---|---|---|---|---|---|---|---|---|---|
| Cliff | BR | 2600.0 | 43.82 | 32.89 | -4.09 | 12.22 | 42.34 | 71.42 | 100.00 |
| | MCE | 2600.0 | 44.49 | 32.88 | -14.42 | 12.89 | 44.38 | 71.84 | 100.00 |
| Gridworld | BR | 2600.0 | 30.95 | 30.25 | -19.69 | 3.23 | 21.74 | 53.54 | 100.00 |
| | MCE | 2600.0 | 28.83 | 28.23 | -20.57 | 3.70 | 21.06 | 46.45 | 99.99 |
| Path | BR | 2600.0 | 40.52 | 34.25 | -60.02 | 7.12 | 37.17 | 67.93 | 100.00 |
| | MCE | 2600.0 | 41.17 | 35.01 | -62.37 | 7.64 | 36.09 | 70.00 | 100.00 |
| RandomMDP | BR | 5320.0 | 10.09 | 19.47 | -42.17 | 0.00 | 0.10 | 14.32 | 99.84 |
| | MCE | 5320.0 | 10.36 | 19.67 | -42.96 | 0.00 | 0.13 | 14.94 | 99.84 |
| TreeMDP | BR | 1920.0 | 21.16 | 26.83 | -44.59 | 0.11 | 9.14 | 38.92 | 100.00 |
| | MCE | 1920.0 | 20.11 | 25.95 | -48.52 | 0.08 | 9.56 | 35.61 | 100.00 |

Table 1: A full breakdown of the true reward lost due to early stopping, with respect to the type of environment and training method used. See Section 3 for the descriptions of environments and reward sampling methods. 320 missing datapoints are cases where numerical instability in our early stopping algorithm implementation resulted in NaN values.

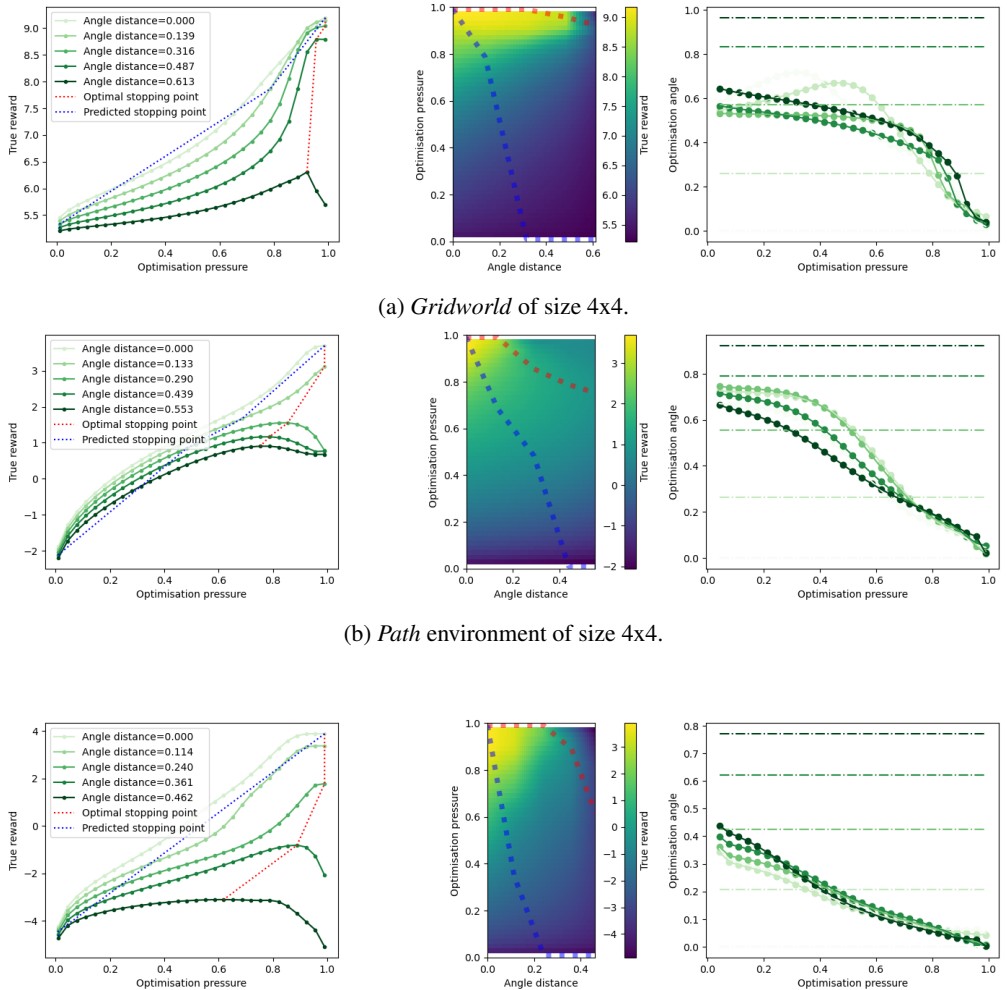

(a) *Gridworld* of size 4x4.

(b) *Path* environment of size 4x4.

(c) *Cliff* environment of size 4x4, with a probability of slipping=0.5.

Figure 9: (Cont. below)

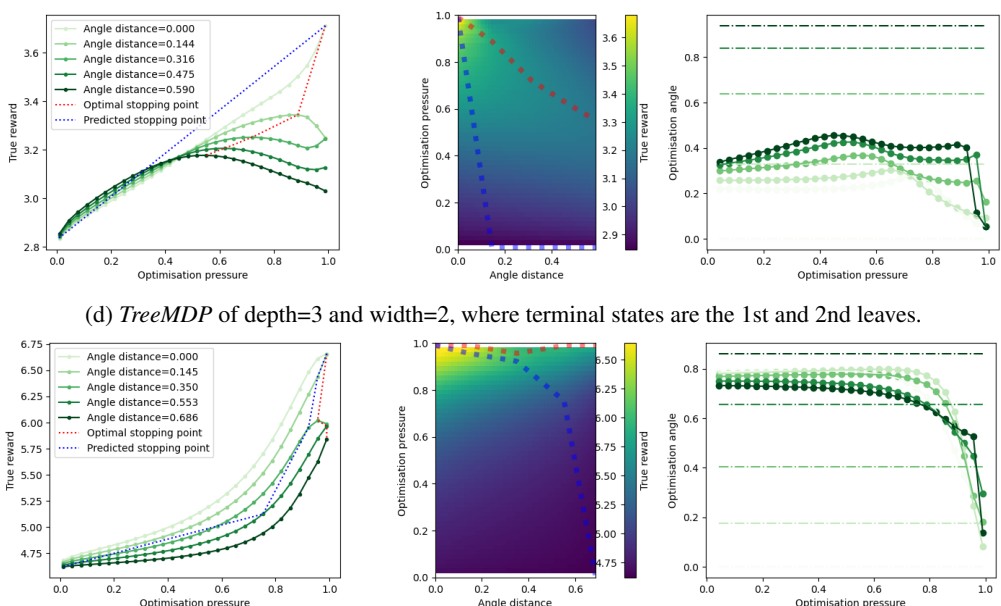

(d) *TreeMDP* of depth=3 and width=2, where terminal states are the 1st and 2nd leaves.

(e) *RandomMPD* of size=16, with 2 terminal states, and 3 actions.

Figure 9: Example runs of the Early Stopping algorithm on different kinds of environments. The left column shows the true reward obtained by training a policy on different proxy rewards, under increasing optimisation pressures. The middle column depicts the same plot under a different spatial projection, which makes it easier to see how much the optimal stopping point differs from the pessimistic one recommended by the Early Stopping algorithm. The right column shows how the optimisation angle (cosine similarity) changes over increasing optimisation pressure for each proxy reward (for a detailed explanation of this type of plot, see Appendix I).

# E A SIMPLE EXAMPLE OF GOODHARTING

In Section 4, we motivated our explanation of Goodhart's law using a simple MDP $\mathcal{M}_{2,2}$, with 2 states and 2 actions, which is depicted in Figure 10. We assumed $\gamma = 0.9$, and uniform initial state distribution $\mu$.

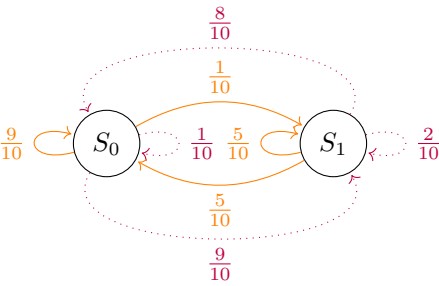

Figure 10: MDP $\mathcal{M}_{2,2}$ with 2 states and 2 actions. Edges corresponding to the action $a_0$ are orange and solid, and edges corresponding to $a_1$ are purple and dotted.

We have sampled three rewards $R_0, R_1, R_2 : S \times A \to \mathbb{R}$, implicitly assuming that $R(s, a, s') = R(s, a, s'')$ for all $s', s'' \in S$.

| $R_0$ | $a_0$ | $a_1$ |
|-------|-------|-------|
| $S_0$ | 0.170 | 0.228 |
| $S_1$ | 0.538 | 0.064 |

(a) Reward 0

| $R_1$ | $a_0$ | $a_1$ |
|-------|-------|-------|
| $S_0$ | 0.248 | 0.196 |
| $S_1$ | 0.467 | 0.089 |

(b) Reward 1

| $R_2$ | $a_0$ | $a_1$ |
|-------|-------|-------|
| $S_0$ | 0.325 | 0.165 |
| $S_1$ | 0.396 | 0.114 |

(c) Reward 2

Figure 11: Reward tables for $R_0, R_1, R_2$

We used 30 equidistant optimisation pressures in $[0.01, 0.99]$ for numerical stability. The hyperparameter $\theta$ for the value iteration algorithm (used in the implementation of MCE) was set to 0.0001.

# F  ITERATIVE IMPROVEMENT

One potential application of Theorem 1 is that when we have a computationally expensive method of evaluating the true reward $R_0$, we can design practical training regimes that provably avoid Goodharting. Typically training regimes for such reward functions involve iteratively training on a low-cost proxy reward function and fine-tuning on true reward using human feedback (Paulus et al., 2018). We can use true reward function to approximate $\arg(R_0, R_1)$, and then optimal stopping gives the optimal amount of time before a "branching point" where possible reward training curves diverge (thus, creating Goodharting).

Specifically, let us assume that we have access to an oracle $\text{ORACLE}_{R^\star}(R_i, \theta_i)$ which produces increasingly accurate approximations of some true reward $R^\star$: when called with a proxy reward $R_i$ and a bound $\theta_i > \arg(R^\star, R_i)$, it returns $R_{i+1}$ and $\theta_{i+1}$ such that $\theta_{i+1} > \arg(R^\star, R_{i+1})$ and $\lim_{i \to \infty} \theta_i = 0$. Then algorithm 1 is an iterative feedback algorithm that avoids Goodharting and terminates at the optimal policy for $R^\star$.

---

**Algorithm 1** Iterative improvement algorithm

---

1: **procedure** ITERATIVEIMPROVEMENT($S, A, \tau$)
2:   $R \sim Unif[\mathbb{R}^{S \times A}]$
3:   $\pi \leftarrow Unif[\mathbb{R}^{S \times A}]$
4:   $\vec{\eta}_{-1} = \vec{\eta}_0 \leftarrow \eta^\pi$
5:   $\theta \leftarrow \frac{\pi}{2}$
6:   $\vec{t}_0 \leftarrow \text{argmax}_{\vec{t} \in T(\vec{\eta}_0)} \vec{t} \cdot R$
7:   **while** $\vec{t}_i \neq \vec{0}$ **do**
8:    **while** $\vec{\eta}_i \cdot \vec{t}_i \leqslant \theta$ **do**
9:     $R, \theta \leftarrow \text{ORACLE}_{R^\star}(R, \theta)$
10:     $R \leftarrow R$
11:    **end while**
12:    $\lambda \leftarrow \max\{\lambda : \vec{\eta}_i + \lambda \vec{t}_i \in \Omega\}$
13:    $\vec{\eta}_{i+1} \leftarrow \vec{\eta}_i + \lambda \vec{t}_i$
14:    $\vec{t}_{i+1} \leftarrow \text{argmax}_{\vec{t} \in T(\vec{\eta}_{i+1})} \vec{t} \cdot R$
15:    $i \leftarrow i + 1$
16:   **end while**
17:   **return** $\eta^{\tilde{\eta}_i - 1}$
18: **end procedure**

---

**Proposition 6.** *Algorithm 1 is a valid optimisation procedure, that is, it terminates at the policy $\pi^\star$ which is optimal for the true reward $R^\star$.*

*Proof.* By Theorem 1, the inner loop of the algorithm maintains that $\mathcal{J}_{R_0}(\pi_{i+1}) \geqslant \mathcal{J}_{R_0}(\pi_i)$. If the algorithm terminates, then it must be that $\vec{t}_i = 0$, and the only point that this can happen is in a point $\eta^{\pi^\star}$.

Since steepest ascent terminates, showing algorithm 1 terminates reduces to showing we only make finitely many calls to $\text{ORACLE}_{R^\star}$. It can be shown that for any $\vec{t}_i$ produced by steepest ascent on $R$, $\vec{t}_i = proj_P(R)$ for some linear subspace $P$ on the boundary of $\Omega$ formed by a subset of boundary conditions. Since there are finitely many such $P$, there is some $\epsilon$ so that for all $R, R^\star$ with $\arg(R, R^\star) < \epsilon$, $\arg(proj_P(R), proj_P(R^\star)) < \frac{\pi}{2}$ for all $P$.

Because we have assumed that $\lim_{i \to \infty} \theta_i = 0$, $\lim_{i \to \infty} \arg(R_i, R^\star) = 0$ and $\text{ORACLE}_{R^\star}$ will only be called until $\arg(R_i, R^\star) = 0$. Then the number of calls is finite and so the algorithm terminates. $\square$

We expect this algorithm forms theoretical basis for a training regime that avoids Goodharting and can be used in practice.

## G    FURTHER EMPIRICAL INVESTIGATION OF GOODHARTING

### G.1    ADDITIONAL PLOTS FOR THE GOODHARTING PREVALENCE EXPERIMENT

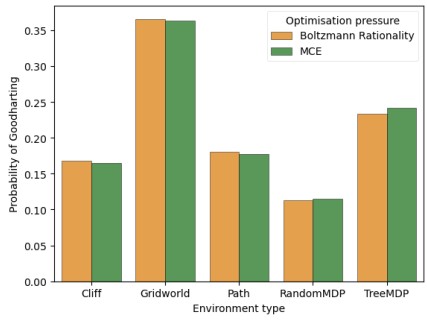
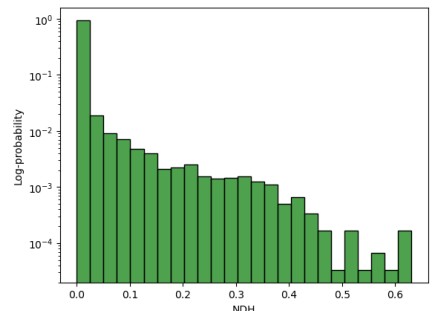

(a)  The  relationship  between  the  type  of
environment and the probability of Goodharting.

(b) Log-scale histogram of the distribution of **NDH
metric** in the dataset.

Figure 12: Summary of the experiment described in Section 3. (a) The choice of operationalisation of
optimisation pressure does not seem to change results in any significant way. (b) NDH metric follows
roughly exponential distribution when restricted to cases when NDH > 0.

### G.2    EXAMINING THE IMPACT OF KEY ENVIRONMENT PARAMETERS ON GOODHARTING

To further understand the conditions that produce Goodharting, we investigate the correlations
between key parameters of the MDP, such as the number of states or temporal discount factor $\gamma$, and
NDH. Doing it directly over the dataset described in Section 3 does not yield good results, as there is
not enough data points for each dimension, and there are confounding cross-correlations between
different parameters changing at the same time.

To address those issues, we opted to replace the grid-search method that produced the datasets
for Section 3 and  Section 5.1.  We have first picked a base distribution over representative
environments, and then created a separate dataset for each of the key parameters, where only
that parameter is varied.[3]

Specifically, the base distributions is given over *RandomMDP* with $|S|$ sampled uniformly between
$8$ and $64$, $|A|$ sampled uniformly between $2$ and $16$, and the number of terminal states sampled
uniformly between $1$ and $4$, where $\gamma = 0.9$, $\sigma = 1$, and where we use 25 $\lambda$'s spaced equally (on
a log scale) between $0.01$ and $0.99$. Then, in each of the runs we modify this base distribution of
environments along a single axis, such as sampling $\gamma$ uniformly across $(0, 1)$ shown in Figure 16c.

**Key findings (positive):**    The number of actions $|A|$ seems to have a suprisingly significant impact
on NDH (Figure 15).  The further away is the proxy reward function from the true reward, the
more Goodharting is observed (Figure 16a), which corroborates the explanation given at the end
of Section 4. We note that in many examples (for example in Figure 2 or in any of graphs in Figure 9)
the closer the proxy reward is, the later the Goodhart "hump" appeared - this positive correlation
is presented in Figure 16b. We also find that Goodharting is less significant in the case of myopic
agents, that is, there is a positive correlation between $\gamma$ and NDH (Figure 16c). Type of environment
seems to have impact on observed NDH, but note that this might be a spurious correlation.

**Key findings (negative):**    The number of states $|S|$ does not seem to significantly impact the amount
of Goodharting, as measured by NDH (Figures 13 and 14). This suggests that having proper methods
of addressing Goodhart's law is important as we scale to more realistic scenarios, and also partially
explains the existence of the wide variety of literature on reward gaming (see Section 1.1).  The
determinism of the environment (as measured by the Shannon entropy of the transition matrix $\tau$)
does not seem to play any role.

---

[3]Since this is impossible when comparing between environment types, we use the original dataset
from Section 3 in Figure 16f.

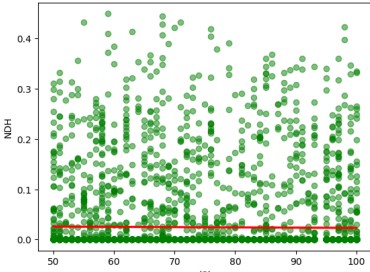 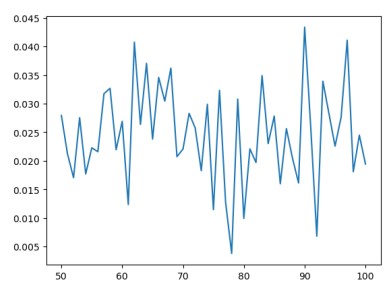

Figure 13: Small negative correlation (not statistically sifnificant) between |S| and NDH: $y = -5e - 05x + 0.02852$, 95% CI for slope: $[-0.00018, 7.23e - 05]$, 95% CI for intercept: $[0.01892, 0.03813]$, Pearson's $r = -0.0119$, $R^2 = 0.0$, $p = 0.4058$. On the left: scatter plot, with the least-squares regression fitted line shown in red. On the right: mean NDH per |S|, smoothed with rolling average, window size=10. **Below, we have repeated the experiment for larger |S| to investigate asymptotic behaviour.** N=1000.

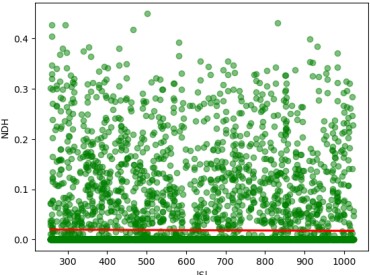 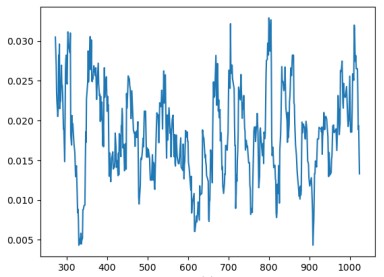

Figure 14: Small negative correlation (not statistically significant) between the number of states in the environment and NDH: $y = -0.0x + 0.02047$, 95% CI for slope: $[-8.2529e - 06, 2.2416e - 06]$, 95% CI for intercept: $[0.017, 0.0239]$, Pearson's $r = -0.0116$, $R^2 = 0.0$, $p = 0.261$. On the left: scatter plot, with the least-squares regression fitted line shown in red. On the right: mean NDH per |S|, smoothed with rolling average, window size=10. N=1000.

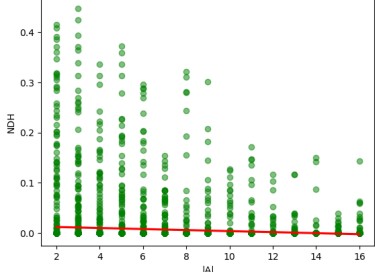 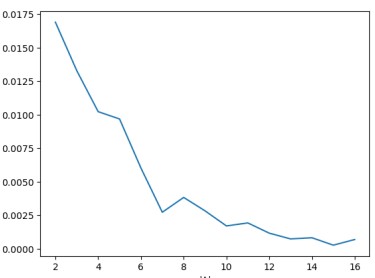

Figure 15: Correlation between |A| and NDH: $y = -0.00011x + 0.00871$, 95% CI for slope: $[-0.00015, -7.28e - 05]$, 95% CI for intercept: $[0.00723, 0.01019]$, Pearson's $r = -0.0604$, $R^2 = 0.0$, $p < 1e - 08$. On the left: scatter plot, with the least-squares regression fitted line shown in red. On the right: mean NDH per |A|. N=1000.

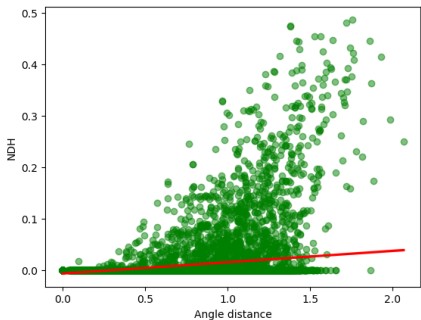

(a) Correlation between the angle distance to the proxy reward and **NDH metric**: $y = 0.02389x - 0.00776$, 95% CI for slope: $[0.02232, 0.02546]$, 95% CI for intercept: $[-0.00883, -0.00670]$, Pearson's $r = 0.2895$, $R^2 = 0.08$, $p < 1.2853e - 187$.

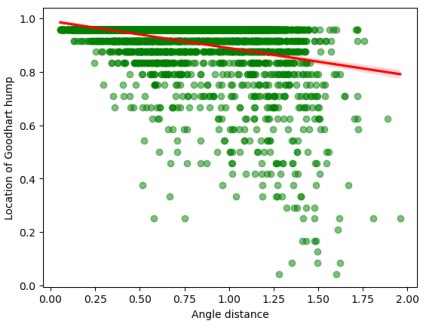

(b) Correlation between the distance to the proxy reward, and the **location of the Goodhart's hump**: $y = -0.10169x + 0.99048$, 95% CI for slope: $[-0.11005, -0.09334]$, 95% CI for intercept: $[0.98360, 0.99736]$, Pearson's $r = -0.3422$, $R^2 = 0.12$, $p < 2.7400e - 118$.

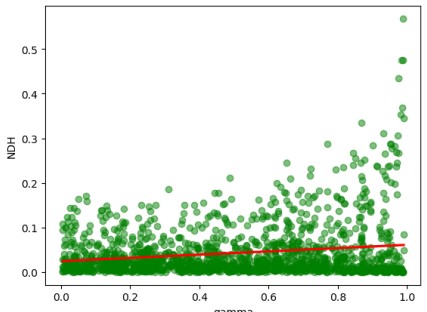

(c) Correlation between $\gamma$ and the **NDH metric**: $y = 0.00696x + 0.00326$, 95% CI for slope: $[0.00504, 0.00888]$, 95% CI for intercept: $[0.00217, 0.00436]$, Pearson's $r = 0.07137$, $R^2 = 0.01$, $p < 1.2581e - 12$.

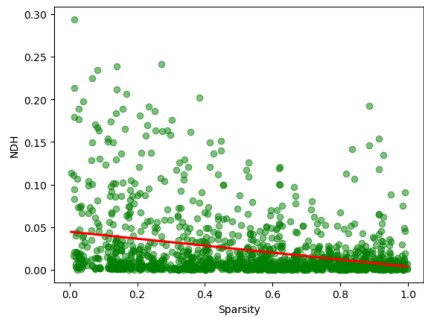

(d) Correlation between the sparsity of the reward, and the **NDH metric**: $y = -0.00701x + 0.00937$, 95% CI for slope: $[-0.00852, -0.00550]$, 95% CI for intercept: $[0.00850, 0.01024]$, Pearson's $r = -0.09090$, $R^2 = 0.01$, $p < 9.5146e - 20$.

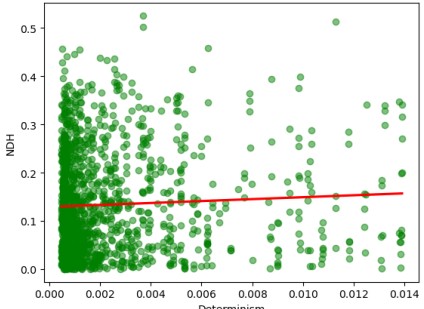

(e) Correlation between the determinism of the environment, and the **NDH metric**: $y = 0.77221x + 0.01727$, 95% CI for slope: $[0.31058, 1.23384]$, 95% CI for intercept: $[0.01570, 0.01884]$, Pearson's $r = 0.03278$, $R^2 = 0.01$, $p = 0.0010$.

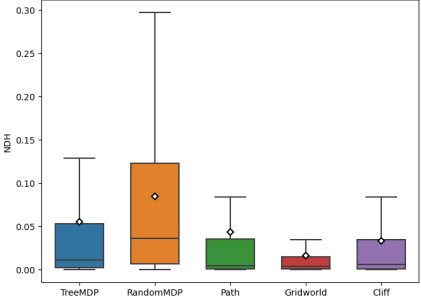

(f) Distribution of NDH metric for different kinds of environments. Note that other parameters are *not* kept constant across environments, which might introduce cross-correlations.

Figure 16: Correlation plots for different parameters of MDPs. N=1000 for all graphs above except for Figure 16f, which uses the dataset from Section 3 where N=30400.

# H  IMPLEMENTING THE EXPERIMENTS

## H.1  COMPUTING THE PROJECTION MATRIX

For reward $R$, we want to find its projection $M_\tau R$ onto the $|S|(|A| - 1)$-dimensional hyperplane $H = \text{span}(\Omega)$ containing all valid policies. $H$ is defined by the linear equation $A\vec{x} = b$ corresponding to the constraints defined in $appendix\ B$, giving $M_\tau = I - A^t(AA^t)^{-1}A$ by standard linear algebra results. However, this is too computationally expensive to compute for environments with a high number of states.

There is another potential method that we designed but did not implement. It can be shown that the subspace of vectors orthogonal to $H$ corresponds exactly to expected reward vectors generated by potential functions - that is, the set of vectors orthogonal to $H$ is exactly the vectors

$$R(s, a) = \mathbb{E}_{s' \sim \tau(s,a)}[\gamma\phi(s')] - \phi(s)$$

for potential function $\phi : S \to \mathbb{R}$. Note this also gives that all vectors of shaped rewards have the same projection, so we aim to shape rewards to be orthogonal to all vectors described above.

To do this, we initialise two potential functions $\phi, \tilde{\phi}$ and consider the expected reward vectors of

$$R_\|(s, a) := R(s, a) + \mathbb{E}_{s' \sim \tau(s,a)}[\gamma\phi(s')] - \phi(s)$$

and

$$R_\perp(s, a) = \mathbb{E}_{s' \sim \tau(s,a)}[\gamma\tilde{\phi}(s')] - \tilde{\phi}(s).$$

We optimise $\phi$ to maximize the dot product between these vectors and $\tilde{\phi}$ to minimize it. $\phi$ converges so that $R_\|$ is orthogonal to all reward vectors $R_\perp$, and will thus be $R$'s projection onto $H$.

## H.2  COMPUTE RESOURCES

We performed our large-scale experiments on AWS. Overall, the process took about 100 hours of a *c5a.16xlarge* instance with 64 cores and 128 GB RAM, as well as about 100 hours of *t2.2xlarge* instance with 8 cores and 32 GB RAM.

## I  AN ADDITIONAL EXAMPLE OF THE PHASE SHIFT DYNAMICS

In the Figure 4, Figure 3 and Appendix E, we have explored an example of 2-state, 2-action MDP. The image space being 2-dimensional makes the visualisation of the run easy, but the disadvantage is that we do not get to see multiple state transitions. Here, we show an example of a 3-state, 2-action MDP, which does exhibit multiple changes in the direction of the optimisation angle.

We use an an MDP $\mathcal{M}_{3,2}$ defined by the following transition matrix $\tau$:

|       | $a_0$ | $a_1$ |
|-------|-------|-------|
| $S_0$ | 0.9   | 0.1   |
| $S_1$ | 0.1   | 0.9   |
| $S_2$ | 0.0   | 0.0   |

(a) *Starting from $S_0$*

|       | $a_0$ | $a_1$ |
|-------|-------|-------|
| $S_0$ | 0.1   | 0.9   |
| $S_1$ | 0.9   | 0.1   |
| $S_2$ | 0.0   | 0.0   |

(b) *Starting from $S_1$*

|       | $a_0$ | $a_1$ |
|-------|-------|-------|
| $S_0$ | 0.0   | 0.0   |
| $S_1$ | 0.0   | 0.0   |
| $S_2$ | 1.0   | 1.0   |

(c) *Starting from $S_2$*

with $N = 5$ different proxy functions interpolated linearly between $R_0$ and $R_1$. We use 30 optimisation strengths spaced equally between 0.01 and 0.99.

|       | $a_0$ | $a_1$ |
|-------|-------|-------|
| $S_0$ | 0.290 | 0.020 |
| $S_1$ | 0.191 | 0.202 |
| $S_2$ | 0.263 | 0.034 |

(a) Reward $R_0$

|       | $a_0$ | $a_1$ |
|-------|-------|-------|
| $S_0$ | 0.263 | 0.195 |
| $S_1$ | 0.110 | 0.090 |
| $S_2$ | 0.161 | 0.181 |

(b) Reward $R_1$

Figure 18: Reward Tables

The rest of the hyperparameters are set as in the Appendix E, with the difference that we are now using exactly steepest ascent with early stopping, as described in Figure 4b, instead of MCE approximation to it.

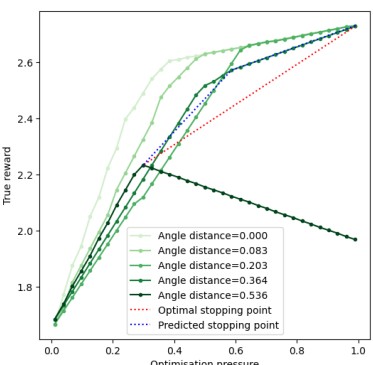 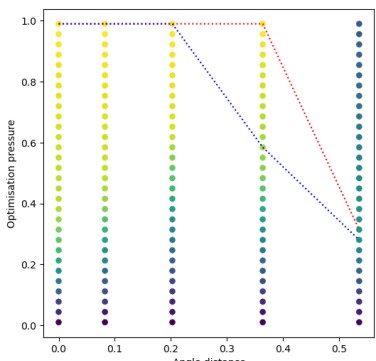

(a) Goodharting behavior for $\mathcal{M}_{3,2}$ over five reward functions. Observe that the training method is concave, in accordance with Proposition 3. Compare to the theoretical explanation in Appendix A, in particular to the figures showing piece-wise linear plots of obtained reward over increasing optimisation pressure.

(b) The same plot under a different spatial projection, which makes it easier to see how much the optimal stopping point differs from the pessimistic one recommended by the Early Stopping algorithm.

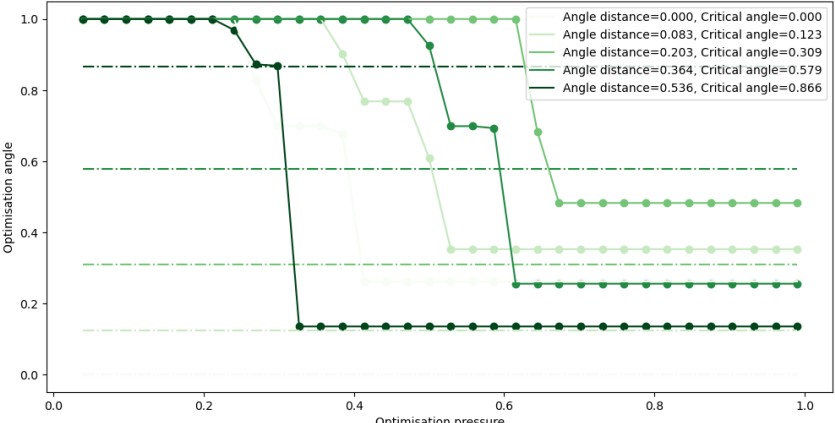

(c) A visualisation of how the optimisation angle (cosine similarity) changes over increasing optimisation pressure for each proxy reward. This is the angle between the current direction of optimisation in $\Omega$, i.e. $(\eta^{\pi_{\alpha_{i+1}}} - \eta^{\pi_{\alpha_i}})$, and the proxy reward function projection $M_\tau R_i$ (defined as $\cos\left(\arg\left(R', \vec{d}\right)\right)$ in the proof of Theorem 1). Once the angle crosses the critical threshold, the algorithm stops. The critical threshold depends on the distance $\theta$ between the proxy and the true reward, and it is drawn in as a dotted line, with a color corresponding to the color of the proxy reward. Compare this plot to Figure 20 - we can see exact places where the phase transition happens, as the training run meets the boundary of the convex space. Also, compare to Figure 19a, where it can be seen how the algorithm stops (in blue) immediately after the training run crosses the corresponding critical angle value (in case fo the last two proxy rewards), or continues to the end (in case of the first two).

Figure 19: Summary plots for the Steepest Ascent training algorithm over five proxy reward functions.

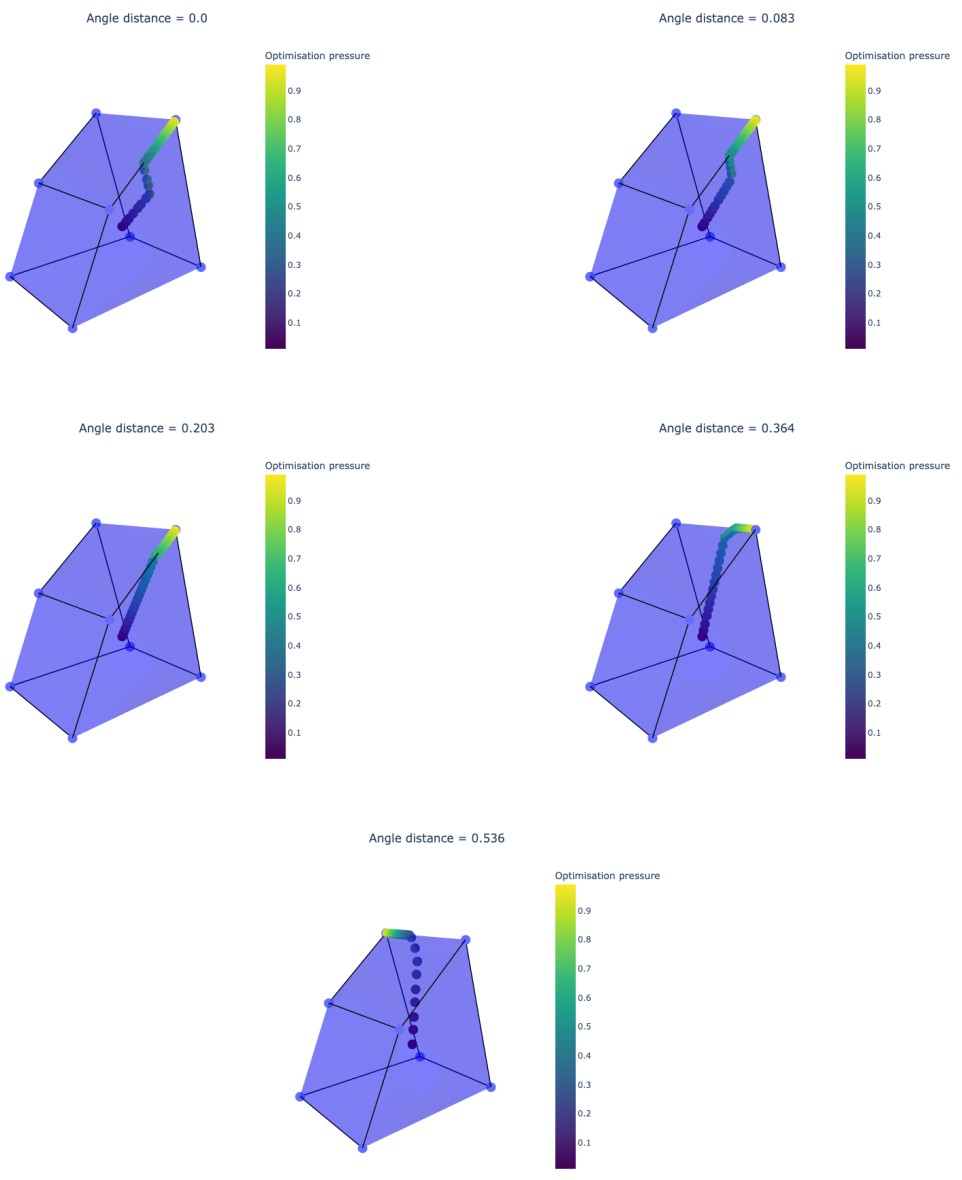

Figure 20: Trajectories of optimisations for using different proxy rewards. Note that the occupancy measure space is $|S|(|A|-1) = 3$-dimensional in this example, and the convex hull is over $|A|^{|S|} = 8$ deterministic policies. We hide the true/proxy reward vectors for presentation clarity.

