# OpenReview forum: "Goodhart's Law in Reinforcement Learning"
_ICLR.cc/2024/Conference — ICLR 2024 poster_

### Official Review · Reviewer_MUg9 · 2023-10-29

**Soundness:** 3 good
**Presentation:** 3 good
**Contribution:** 3 good
**Rating:** 8
**Confidence:** 2

**Summary:**

This paper studies the interplay between reward misspecification and optimisation in RL as an instance of Goodhart's law. The authors provice geometric explanations of how optimisation of a misspecified reward function can lead to worse performance beyond some threshold, and show in experiments that several environments and reward functions are prone to Goodhart's law and optimisation of a reward proxy eventually leads to worse performance. The authors also propose an early stopping algorithm to address this problem.

**Strengths:**

- First of all, the studied topic is, in my opinion, important and could be of interest to many in the ICLR community.
- The paper is a good attempt at extending prior work on reward misspecification and reward gaming (e.g., Skalse et al. 2022) to the question of what role optimisation plays and whether we can characterize reward misspecification from a policy optimisation standpoint as well. I am not very well acquainted with the related work, but the contributions and many of the ideas in this paper seem novel to me.
- The results are very interesting and provide some nice intuition about the interplay of reward distance, optimisation and MDP model. While I don't think that one should overinterpret the results as they are either based on empirical studies of a some specific set of environments or on theoretical insights with idealised assumptions, I think that the findings of this paper are overall very interesting.

**Weaknesses:**

- The evidence on the "Goodharting" effect are only circumstantial. Experiments on some specific set of environments such as grid worlds do not necessarily allow us to extrapolate. After all, the Goodharting effect can only be "explained" but not characterised. Nevertheless, these experiments and the geometric explanations provide good intuition which I think is very interesting and could inspire future lines of work.
- A minor weakness is that the proposed early stopping algorithm might not perform well due to large reward losses from stopping early, which is somewhat expected due to its pessimistic nature. The algorithm is also fairly impractical bcause it assumes prior knowledge of $\theta$.

**Questions:**

- Your work seems to be tailored to the specific choice of difference metric between two reward functions (their angle). I guess that the main reason for choosing this distance metric is that it is a STARC metric.
	- However, can you provide further justification for why the "angle" is a good choice or even the *right* choice?
	- What could another reasonable metric be?
	- And, how would choosing a different metric impact your results?

---

> ### Author Response · Authors · 2023-11-20
> **Response**
>
> We kindly thank you for your review.
>
>  - *A minor weakness is that the proposed early stopping algorithm might not perform well due to large reward losses from stopping early, which is somewhat expected due to its pessimistic nature.* It is true that the pessimistic nature of the algorithm leads to the loss of reward. However, as we point out in Section 5 (Figure 5b), if the proxy is not too far from the true reward, then one can probably expect the loss of reward to be fairly small. However, we think that In practice, early stopping algorithm will not be used on its own, but rather as part of the iterative improvement algorithm (described in Appendix F). This is how our work might lead to some sort of improved RLHF training regime that provably improves Goodharting. (We can then make calls to the true reward function through human feedback to estimate theta, and conservative stopping just translates to collecting human feedback throughout the training process, rather than all at once - which is also how RLHF training processes currently look).
>
>  - *The algorithm is also fairly impractical because it assumes prior knowledge of θ.* We argue that knowing the upper bound of the distance between the true and proxy reward functions is not an extremely unrealistic assumption. For example, for a given *reward learning algorithm,* we might have certain bounds or rates of convergence to the true reward. Inverse reinforcement learning algorithms often come with proofs of their convergence rates - see e.g. "Maximum-Likelihood Inverse Reinforcement Learning with Finite-Time Guarantees”, Zeng et al., 2022. Using those convergence rates, we could get an approximation of the distance to the true reward. Note that we can always make the bound less tight, and therefore trade off the accuracy of the input to the early stopping and the optimality of the stopping point, which means that even in the presence of a weak signal our approach might still be useful.
>
>  - *Your work seems to be tailored to the specific choice of difference metric between two reward functions (their angle). I guess that the main reason for choosing this distance metric is that it is a STARC metric. However, can you provide further justification for why the "angle" is a good choice or even the right choice?* We have actually developed the angle metric independently, and only later realised that it is indeed a STARC metric. When working in the occupancy measure space, it is natural to consider the angle between the (projected) rewards - the dot product arises naturally when considering the distances between (normalised) vectors in the Euclidian space. We also note that Skalse et al. 2023, “Invariance in Policy Optimisation and Partial Identifiability in Reward Learning” (indirectly) show that many reward learning algorithms will converge to a reward with STARC-distance 0 to the true reward.
>
>    - *What could another reasonable metric be?* Our angle metric can be understood as an expectation $E_{S \sim U, A \sim U}[R(S, A) R(S, A)]$ , where both states and actions come from the uniform distribution. It would be interesting to develop a better understanding of the metric of the shape $ E_{S,A \sim D}[R(S,A), R[S, A])$ for a larger class of distributions $D$ over states and actions. Particularly interesting are the distributions induced by a particular policy $D_\pi$. Note that this is, in a way, no longer independent of where we are on the polytope $\Omega$, which presents more conceptual difficulties.
>
>    - *And, how would choosing a different metric impact your results?* As long as we work with a STARC metric, we can always rescale the by a strictly monotonic function and get back to the proper angle. If we use a different distance, our geometric intuition might no longer hold.

---

> > ### Comment · Reviewer_MUg9 · 2023-11-22
> > **Thanks for the response**
> >
> > Thank you for your response. Reading the other reviewers' comments and briefly going through the paper again, I've become a bit unsure whether the results of Section 5 are as significant as I originally deemed them to be. It seems that a statement such as in Corollary 1 could also be derived without much additional work from well-known results like the simulation lemma, which is anyways quite similar I guess. Nevertheless, I am still in favour of acceptance, though, I am reducing my confidence.

---

> > > ### Author Response · Authors · 2023-11-23
> > > **Response**
> > >
> > > We appreciate the feedback, and giving us a chance to clarify. We claim that the simulation lemma, although concerned with the topic of MDP approximation, does not capture our contribution, nor allows it to be easily deduced.
> > >
> > > Specifically, simulation lemma (as given in [1]) assumes that we have an MDP\R $M$ and a pair of reward functions $R, R'$ with means $R_\mu, R_\mu'$ such that $||R_\mu - R_\mu'||_\infty < \epsilon$. Then, it allows to deduce that returns of the optimal policies on $(M, R)$ and$(M, R')$ are also sufficiently close (bounded polynomially in the relevant quantities).
> > >
> > > It is true that our work is situated in a similar setup: we have an MDP\R $M$ and a pair of reward functions $R, R'$. We then assume that $\text{angle}(R, R') < \epsilon$, where $\text{angle}(\cdot, \cdot)$ is the distance between reward given by the angle of their projections onto the relevant subspace in the occupancy measure space. From then, in the Corollarly 1, given a starting policy (again, given as a point in the occupancy measure space), and some concave (in the relevant sense) learning algorithm, we prove that the optimal stopping point of training using this algorithm is given by the
> > >
> > > $$
> > > \frac{J_R(\pi_{i+1}) - J_R(\pi_i)}{||\eta^{\pi_{i+1}} - \eta^{\pi_{i+1}}||} \leq \sin(\theta)||M_\tau R||
> > > $$
> > >
> > > where $\pi_1, \pi_2,\dots$ is the sequence of policies obtained from that algorithm.
> > >
> > > Apart from being concerned about a similar problem, our formalisation and the whole setup is significantly different. The dynamics of training given the geometry of the space of policies is not considered at all in [1]. Nor are the properties of the space of rewards, which are crucial to understand why does the angle is a better metric than $\infty$-norm. We don’t see how could one easily deduce our result from this prior work.
> > >
> > > Note that the simulation lemma’s bounds do not change for different policies. This makes it impossible to derive any stopping algorithms from it, as such algorithms would need to account for how returns differ for different policies, as well as training dynamics.
> > >
> > > The simulation lemma assumes the reward functions are very close; they assume that for every state and action the reward functions are within $\epsilon$. This does not meaningfully capture Goodhart’s law issues, which occur when reward functions disagree on a few state/action pairs, so value functions are high only for specific distributions. For example, cheating on tests clearly is not within $\epsilon$ for the reward functions of *GPA* and measuring *learning outcomes* directly. If value functions are close for all policies, in some sense a lot of our work in the paper would be meaningless since optimal stopping is particularly useful in cases where reward decreases drastically for policies that perform too well on the proxy reward.
> > >
> > > Thank you again for giving us the chance to address your points of uncertainty, we hope that it made our contribution clearer now.
> > >
> > > [1] Kearns, M., Singh, S. Near-Optimal Reinforcement Learning in Polynomial Time. *Machine Learning* **49**, 209–232 (2002). https://doi.org/10.1023/A:1017984413808

---

### Official Review · Reviewer_eHjA · 2023-11-01

**Soundness:** 4 excellent
**Presentation:** 4 excellent
**Contribution:** 3 good
**Rating:** 6
**Confidence:** 3

**Summary:**

Most reinforcement learning algorithms are designed for accurate reward feedback. However, in practice, accurate reward feedback may not be available. In the presence of inaccurate reward feedback, it is possible to observe a phenomenon that the performance of the training policy first increases and then decreases after passing a threshold point. This paper addresses this interesting phenomenon and names it “Goodhart’s Law in RL”. To solve this problem, this paper quantifies the magnitude of this effect and how it exists in a wide range of environments and reward functions. It provides a geometric explanation and an optimal early stopping method with theoretical regret bounds. They then empirically showed the performance of their early stopping method.

**Strengths:**

1. This paper is quite novel because it raises an interesting and important observation – the performance of a policy increases first and then decreases. Such observation is caused by inaccurate reward feedback, which indeed exists in real RL applications.

2. This paper quantifies the magnitude of such phenomena and provides a clear geometric explanation.

3. With these insights, this paper proposes an optimal early stopping method with theoretical regret bound analysis.

4. The experimental results supported the authors' claim.

5. This paper is well-written. Concepts are conveyed efficiently. The analysis is detailed while keeping a clear line of high-level logic.

**Weaknesses:**

1. The optimal early stopping rule relies on the knowledge of the occupancy measure and the upper bound $\theta$ of the angle between the true reward and the proxy reward. Methods to approximate the occupancy measure are well-researched. My concern is on the approximation of $\theta$, which is a relatively new concept and requires some knowledge of the true reward feedback or true reward samples. When such estimation is not accurate, the stopping method could exhibit negative performance. It would be better if the author could show empirical results with approximated $\theta$.

2. This paper is preliminary because it only considers finite state and action space. The empirical results are also only on small grid world environments. It is not clear whether such a phenomenon exists in more broad continuous settings and what would be the practical way to solve it in these settings.

**Questions:**

N/A

---

> ### Author Response · Authors · 2023-11-20
> **Response**
>
> Thank you very much for your review. Regarding the weaknesses you have pointed out:
>
> - *My concern is on the approximation of θ, which is a relatively new concept and requires some knowledge of the true reward feedback or true reward samples. When such estimation is not accurate, the stopping method could exhibit negative performance. It would be better if the author could show empirical results with approximated θ.* The approximation of the distance between the true and proxy reward functions is not necessarily unprecedented. For example, for a given reward learning algorithm, we might have certain bounds or rates of convergence to the true reward. Inverse reinforcement learning algorithms often come with proofs of their convergence rates - see e.g. "Maximum-Likelihood Inverse Reinforcement Learning with Finite-Time Guarantees”, Zeng et al., 2022. Using those convergence rates, we could get an approximation of the distance to the true reward.
> We think that in practice, the early stopping algorithm will not be used on its own, but rather as a component of the iterative improvement algorithm (discussed in Appendix F). That algorithm provably converges to the optimal policy only if we keep the invariant of never falling into Goodharting. Therefore, using a sufficiently large number of real reward evaluations, in the end we do not lose any reward. In this way, we see our work potentially leading to some sort of improved RLHF training regime. (We can then make calls to the true reward function through human feedback to estimate theta, and conservative stopping just translates to collecting human feedback throughout the training process, rather than all at once - which is also how RLHF training processes currently look).
> We note that our work is meant to be the first step in the investigation of Goodhart’s law in RL: we feel that a proper empirical test of the effects of approximated θ on the performance of the algorithm would benefit from deferring to future work. It would require implementing and testing different reward learning algorithms, which unfortunately lies outside of our scope here.
> - *This paper is preliminary because it only considers finite state and action space.* Extending our theory to the countable or continuous state spaces is an interesting problem. It seems that Theorem 1 holds for countably-infinite state and action spaces, the only change is in the proof of Proposition 1 and Lemma 1. In a continuous setting, the theory becomes more involved, and might require more complex tools from functional analysis. (Note most of our results just rely on separating the policy from the reward, which the above shows we can still do in the general setting.)
> - *The empirical results are also only on small grid world environments. It is only clear whether such a phenomenon exists in more broad continuous settings and what would be the practical way to solve it in these settings.*  We note that we use three fundamentally different types of MDPs (densely-connected, tree-shaped, and some variations of grid worlds).  Our key results being similar across the classes gives us some evidence that they also generalise to more complex environments. Grid worlds constitute a standard evaluation tool for RL algorithms, densely connected MDPs are used in place of “uninformatively-random” environment, while tree-like environments are good models for many game or search problems. We also note that our Goodharting curves match the ones found in Gao et al. 2022, who use much larger state spaces (a LLM model) and much more complex reward models. However, working out how to apply the early stopping algorithm in those large or infinite spaces is indeed an open problem.
>
> We hope that this helps to clarify our paper, and that the reviewer might consider increasing their score.

---

### Official Review · Reviewer_YkV9 · 2023-11-03

**Soundness:** 3 good
**Presentation:** 3 good
**Contribution:** 3 good
**Rating:** 6
**Confidence:** 3

**Summary:**

This paper studies the problem of reward misspecification. The authors point out that over-optimizing an incorrect reward function can lead to the wrong behaviour for the true ("test") reward function, and dub this phenomenon Goodharting. The authors propose a quantitative way to evaluate this phenomenon (cf. Definition 5), and perform an experimental case study on some simple MDPs to establish that Goodharting is a common phenomenon in RL. The authors then provide an intuitive geometric explanation for this phenomenon and propose an early stopping method to avoid overfitting. Further experimental evaluations are performed on the early stopping method to

**Strengths:**

The paper investigates an interesting, albeit not entirely surprising phenomenon, and investigates it thoroughly and carefully. The problem of reward misspecification is quite relevant for practical considerings of RL, so gaining some understanding of this problem is appreciated. The paper is well-written and the messages are conveyed clearly. The theoretical contributions, while not exactly practical, are a nice step towards preventing this problem from affecting performance.

**Weaknesses:**

While I am overall positive about the paper, I have a few comments and suggestions for possible improvement.

- The definition of optimization pressure is a bit strange. Why should we not define it as simply the distance from the optimal policy? For instance, we can say that the optimization pressure is epsilon if we obtain a policy $\hat{\pi}$ such that $J_R(\pi^\star) - J_R(\hat{\pi}) \leq \varepsilon$. I feel that tying the optimization pressure to a certain regularization scheme detracts from the fundamental aspect of the problem, and furthermore that regularization is only used here as a proxy for "how close to optimal are we", which can be defined more directly as above.
- The environments that have been used to establish that Goodharting is pervasive (Section 3) are somewhat simple. I understand that it is difficult to measure the NDH metric in environments where we cannot solve for the optimal policy, but it would have been nice to understand how pervasive this is in "real" problems, or at least in popular RL benchmark environments. As a side note, the fact that the NDH metric is inherently difficult to measure can be considered as a drawback of the proposed methodology -- can the authors comment?
- It would also have been interesting to more systematically study which properties of environments imply that Goodharting is more likely to take place, do the dynamics of the MDP (e.g. a bottleneck structure) have any role?
- The proposed optimal stopping algorithm is very pessimistic since it tries to avoid overfitting to any possible reward function in a certain set (is this pessimism unavoidable?), and as the authors point out it is computationally infeasible. In addition, if I understand correctly, it requires knowing the transition dynamics and knowing the distance between the proxy reward and the true reward function, which is fairly unpractical.

- Incorrect/unclear sentences:
1. "We observe that NDH is non-zero if and only if, over increasing optimisation pressure, the proxy and true rewards are initially correlated, and then become anti-correlated". I believe the authors meant the NDH is non-negative, not non-zero.
2. "Note that this scheme is valid because for any environment, reward sampling scheme and fixed parameters, the sample space of rewards is convex. In high dimensions, two random vectors are approximately orthogonal with high probability, so the sequence R_t spans a range of distances.". It is not clear what point the first sentence is attempting to communicate (what does "valid" mean?), and the second sentence is incorrect as stated (what distribution is one sampling from? I can imagine many distributions where this is untrue, say a deterministic one.)

**Questions:**

See weaknesses section above.

---

> ### Author Response · Authors · 2023-11-20
> **Response**
>
> We are grateful for your detailed comments and questions. We give a detailed answer point-by-point below:
>
> - *Why should we not define it [the optimisation pressure] as simply the distance from the optimal policy?* The optimisation pressure is intended to be an independent variable that we can vary freely, such that increasing the pressure leads to policies progressively closer to optimal (i.e. a parametrisation of some curve between the initial policy and the optimal policy). It is unclear how directly varying the distance $\epsilon$ (such that  $J(\pi) - J(\pi') \leq \epsilon$) could work in practice. The contour lines (inverse images of $\epsilon$) in this case would result in a sequence of n-dimensional spheres in the space of policies, not a linearly ordered sequence of policies.
> - *The environments that have been used to establish that Goodharting is pervasive (Section 3) are somewhat simple.* It is true that we mostly used small environments with at most a couple hundred states, which are far from real-life conditions. However, we do use three fundamentally different types of MDPs (densely-connected, tree-shaped, and some variations of grid worlds). Our key results being similar across the classes gives us some evidence that they also generalise to more complex environments. Grid worlds constitute a standard evaluation tool for RL algorithms, densely connected MDPs are used in place of “uninformatively-random” environment, while tree-like environments are good models for many game or search problems. We also note that our Goodharting curves match the ones found in Gao et al. 2022, who use much larger state spaces (a LLM model) and much more complex reward models.
> - *I understand that it is difficult to measure the NDH metric in environments where we cannot solve for the optimal policy, but it would have been nice to understand how pervasive this is in "real" problems, or at least in popular RL benchmark environments. As a side note, the fact that the NDH metric is inherently difficult to measure can be considered as a drawback of the proposed methodology -- can the authors comment?* We agree that the fact that the NDH metric requires knowledge of the true reward is a drawback. At the same time, we feel that every “complete” measure of Goodhart’s law would require it (”complete” meaning a measure that is non-zero for any non-optimal policy, w.r.t. the true reward). Therefore, we still think it useful to have it defined - and supported by experiments, which we discuss in Appendix C. We expect that in cases where computing true NDH is computationally prohibitive, some approximate methods can be used. In our setup, we did not need to develop such approximations, so we leave this topic for future work. Another point of view might be that even though we used NDH, our reason for choosing it was simplicity (as we note in the paper), since all the measures we investigated were significantly correlated. Therefore, you could read the contribution here as the fact about the correlation of all “sensible” measures, rather than about the primacy of NDH. That gives us a reason to expect that approximating NDH is achievable in practice.
> As a side note, we want to point out that even though the measure is predicated on knowing the true reward, the point of our work is developing a better understanding of how to optimise misspecified reward *without* knowing the true reward. Therefore, the NDH metric is primarily useful for analysis, and its weak applicability in the deployment setting would not be an issue.
> - *It would also have been interesting to more systematically study which properties of environments imply that Goodharting is more likely to take place, do the dynamics of the MDP (e.g. a bottleneck structure) have any role?* We agree that this is an interesting question! We have conducted the preliminary investigation, but due to a lack of space, we have delegated it to Appendix G. Specifically, we look at how different hyper-parameters of the experiment (the kind of the environment, the number of states |S|, number of actions |A|, temporal discount factor $\gamma$, sparsity of the environment, determinism of the environment, and more) correlate with the severity of Goodharting. We agree that investigating more systematically the key properties of the environment that would potentially give a list of “sufficient and necessary” conditions for significant Goodharting to occur (such as the *bottleneck structure on an MDP* you mentioned) would be very interesting, but we feel is outside of the scope of this work.

---

> > ### Author Response · Authors · 2023-11-20
> > **Response**
> >
> > - *The proposed optimal stopping algorithm is very pessimistic since it tries to avoid overfitting to any possible reward function in a certain set (is this pessimism unavoidable?), and as the authors point out it is computationally infeasible.* It is true that the algorithm can be quite pessimistic, for example losing (on average) half of the reward in some environments. One sense in which the pessimism is necessary is when we consider the early stopping as the component of the iterative improvement algorithm (discussed in Appendix F). The algorithm provably converges to the optimal policy only if we keep the invariant of never falling into Goodharting. Therefore, using a sufficiently large number of real reward evaluations, in the end we do not lose any reward! We think that in practice, our work might lead to some sort of improved RLHF training regime. We can then make calls to the true reward function through human feedback to estimate theta, and conservative stopping just translates to collecting human feedback throughout the training process, rather than all at once - which is also how RLHF training processes currently look.
> > Another interesting question is what if we relax the condition and care about the lack of Goodharting only up to some probability. It seems likely that our framework allows this sort of analysis (since, in the end, it boils down to linear programming). However, we feel that investigating this question in depth is better left for subsequent work, as it requires careful consideration of measures on the space of reward functions.
> > - *In addition, if I understand correctly, it requires knowing the transition dynamics and knowing the distance between the proxy reward and the true reward function, which is fairly unpractical.* Note that we do not need to know the distance, only the upper bound on the distance. The worse the bound, the more reward early-stopping loses - knowing the exact distance leads to stopping at the optimal point. Inverse reinforcement learning algorithms often come with proofs of their convergence rates - see e.g. "Maximum-Likelihood Inverse Reinforcement Learning with Finite-Time Guarantees”, Zeng et al., 2022. Using those convergence rates, we could get an approximation of the distance to the true reward.
> > We also note that the angle distance is a STARC metric, and it is similar to the EPIC metric. There has been a considerable amount of literature on using EPIC in practice, demonstrating how to compute the angle alignment of true and proxy reward in concrete settings (e.g. "Skill-Based Reinforcement Learning with Intrinsic Reward Matching”, Adeniji et al. 2023, where they apply EPIC-based learning for robotic control, or "Vision-Language Models are Zero-Shot Reward Models for Reinforcement Learning” Rocamonde et al., 2023, who describe computing EPIC for CLIP-based rewards).
> > - *"We observe that NDH is non-zero if and only if, over increasing optimisation pressure, the proxy and true rewards are initially correlated, and then become anti-correlated". I believe the authors meant the NDH is non-negative, not non-zero.* Thank you, we meant to define NDH the other way around (i.e. as $\max_\lambda J(\pi_\lambda) - J(\pi_1)$), such that it is always non-negative (this is how it is used in the remainder of the paper). We will fix that.
> > - *"Note that this scheme is valid because for any environment, reward sampling scheme and fixed parameters, the sample space of rewards is convex. In high dimensions, two random vectors are approximately orthogonal with high probability, so the sequence $R_t$ spans a range of distances.". It is not clear what point the first sentence is attempting to communicate (what does "valid" mean?), and the second sentence is incorrect as stated (what distribution is one sampling from? I can imagine many distributions where this is untrue, say a deterministic one.)* You are right that in general, this is not true. However, here we are only speaking about the validity of the method for our chosen environments and distributions of rewards (described in Section 3.1) - all of them are convex sets, and for all of them, sampling from the distribution gives approximately orthogonal reward functions. We will rephrase this point to avoid ambiguity.
> >
> > We hope that this clarifies our contributions, and that the reviewer will consider increasing their score if we have managed to address the crux of their critique.

---

### Official Review · Reviewer_Y2A1 · 2023-11-05

**Soundness:** 3 good
**Presentation:** 3 good
**Contribution:** 2 fair
**Rating:** 5
**Confidence:** 3

**Summary:**

This paper presents an analysis of Goodhart’s law in reinforcement learning. The paper starts with a formalization of the Goodhart problem in terms of misalignment of reward functions on finite MDPs: one proxy reward that the policy optimizes when we really wish to optimize the other. The paper justifies that the problem occurs in small scale experiments demonstrating that increasing the optimization pressure on the proxy eventually leads to a decrease in the true reward. A theoretical analysis is given with some examples about why this occurs in finite MDPs. Finally an early-stopping algorithm is proposed to mitigate this issue along with some preliminary experiments on the algorithm.

**Strengths:**

The problem is clearly very important and a better understanding of proxy rewards, overoptimization, and Goodhart’s law are definitely needed in the community.

The paper is presented fairly clearly, except in some areas which I point out later.

The paper provides insights from multiple frontiers to help shape this understanding (empirical, theoretical, and conceptual).

The theoretical findings are useful, but not entirely surprising given what is known already in the literature (see below). However, I do believe it’s useful to have this formalized and characterized when specifically talking about Goodhart’s law.

**Weaknesses:**

My primary complaint is that, although this is a solid analysis, I do not believe it strikes the heart of the Goodhart problem. The position of the paper is that misalignment can be characterized by the worst-case angle between reward functions. This is a fairly well-understood setting (e.g. see ‘simulation lemma’ by Kearns & Singh or any number of classical RL papers). However, it’s unclear how this maps into problems that (1) are beyond the finite case, or (2) are classical examples of Goodhart’s law like the snake bounty. While one could model (2) in the framework studied here, I am not sure this would be an informative model in those settings as the ‘theta’ is just so large.

The above is more of a conceptual disagreement about the premise. For the rest of the review, I give the benefit of the doubt and simply accept the premise is true.

Unfortunately most of the important empirical results have been relegated to the appendix, leaving the main paper with vague / difficult-to-verify statement such as ‘a Goodhart drop occurs for x% of all experiments. Without figures or tables, it’s difficult to understand what this means, such as what the criteria of a ‘Goodhart drop’ is (any non-zero drop, some negligible drop, etc). It would be helpful to make room in the main paper for results that present a more comprehensive picture of the findings.

The early stopping proposal is natural, but also seems very conservative. This appears to be consistent with the empirical findings. Furthermore it requires knowledge of $\theta$, which is just assumed to be known. While it’s hard to imagine anything can be down without some knowledge of the true reward or structure, this seems quite coarse.

Figure 5 is difficult to appreciate in absolute terms as one cannot tell if, for example, 0.4 is a large value relative to the reward achievable. I think this plot would be better replaced with a typical plot showing how the true and proxy rewards change as the policy is optimized and when the algorithm decides to stop, as well as the counterfactual of what would happen if it does not stop.

**Questions:**

How do you think the theoretical results generalize to the setting where the reward function is considerably more sophisticated than simple finite MDPs? For example, high dimensional, continuous state-action, long-horizon problems?

---

> ### Author Response · Authors · 2023-11-20
> **Response**
>
> Thank you for your comments!
>
> - The simulation lemma is indeed somewhat related to our work - specifically regarding the regret bound for our early stopping algorithm. However, we disagree that it already captures our theoretical contribution. The lemma provides the bound on the maximal reward obtained, while we are more concerned about the dynamics of the training, which the lemma does not take into account.
> - It is a fair point that our work does not exhaust all relevant facets of Goodhart's law. Still, we note that the point of view we take already has precedence in the literature, and, for example, informs and explains some of the prior results on optimising LLMs with RLHF (e.g. Gao et al, 2022). Manheim and Gabarrant, 2019 provide a classification of different aspects of Goodhart’s law in ML: our work is best understood as (mostly) addressing the *Regressional Goodharting,* and, in principle, potentially extending to *Extremal Goodharting*.
> - Regarding your specific point about the vagueness of the statement *‘a Goodhart drop occurs for x% of all experiments'. Without figures or tables, it’s difficult to understand what this means, such as what the criteria of a ‘Goodhart drop’ is (any non-zero drop, some negligible drop, etc)"*. We would like to point out that immediately after writing this we clarify: "*meaning that the NDH > 0*".
> As you noted, we are studying the problem from a conceptual, formal, and empirical point of view, which creates tradeoffs in what to include in the main text. We chose the high-level logic of the paper to be the four-step reasoning “What is Goodharting? → Does it occur in practice? → Why does it occur? → How can we prevent it?”.
> Separately, we did investigate many other questions - in particular, the relationship between kinds of environments/ hyperparameters and the severity of Goodharting, and provided the conclusions and all relevant tables and figures Appendix G. It was difficult to fit all this in the main text without sacrificing other parts, which we felt were more important to understand the “critical path” of the argument outlined above.
> - *The early stopping proposal is natural, but also seems very conservative.* It is true that the conservative nature of the early algorithm might lead to a significant loss of reward, when used naively. But even then, as we point out in Section 5 (Figure 5b), if the proxy is not too far from the true reward, then one can probably expect the loss of reward to be fairly small. We have reasons to suspect this will be true in practice: we note that Skalse et al. 2023, “Invariance in Policy Optimisation and Partial Identifiability in Reward Learning” (indirectly) show that many reward learning algorithms will converge to a reward with STARC-distance 0 to the true reward.
> However, potentially an even more important point is that we think in practice early stopping algorithm will not be used on its own, but rather as part of the iterative improvement algorithm (described in Appendix F). This is how our work might lead to some sort of improved RLHF training regime. We can then make calls to the true reward function through human feedback to estimate theta, and conservative stopping just translates to collecting human feedback throughout the training process, rather than all at once - which is also how RLHF training processes currently look.
> - *Furthermore it requires knowledge of θ, which is just assumed to be known.* We argue that knowing the upper bound of the distance between the true and proxy reward functions is not an unrealistic assumption. For example, for a given reward learning algorithm, we might have certain bounds or rates of convergence to the true reward (e.g. "Maximum-Likelihood Inverse Reinforcement Learning with Finite-Time Guarantees”, Zeng et al., 2022). We also note that the angle distance is a STARC metric, and it is similar to the EPIC metric. There has been a considerable amount of literature on using EPIC in practice, demonstrating how to compute the angle alignment of true and proxy reward in concrete settings (e.g. "Skill-Based Reinforcement Learning with Intrinsic Reward Matching”, Adeniji et al. 2023, where they apply EPIC-based learning for robotic control, or "Vision-Language Models are Zero-Shot Reward Models for Reinforcement Learning” Rocamonde et al., 2023, who describe computing EPIC for CLIP-based rewards). The last thing to note is that we can always make the bound less tight, and therefore trade off the accuracy of the input to the early-stopping and the optimality of the stopping point, which means that even in the presence of a weak signal our approach might still be useful.

---

> > ### Author Response · Authors · 2023-11-20
> > **Response**
> >
> > - *Figure 5 is difficult to appreciate in absolute terms as one cannot tell if, for example, 0.4 is a large value relative to the reward achievable. I think this plot would be better replaced with a typical plot showing how the true and proxy rewards change as the policy is optimized and when the algorithm decides to stop, as well as the counterfactual of what would happen if it does not stop. A*ll the reported values in Figure 5 are in percentage points, as indicated in the main text. We will revise the caption of the figure to emphasise this. In that figure, we report the values aggregated over many thousands of runs to support our conclusions (such as the fact our technique works independently of the choice of operationalisation, and that the loss of reward is less significant for small angles theta). Individual runs in the format you describe are presented for illustrative purposes in e.g. Figure 3a, while a bigger collection is delegated to Figure 9 in Appendix D.
> > - *How do you think the theoretical results generalize to the setting where the reward function is considerably more sophisticated than simple finite MDPs? For example, high dimensional, continuous state-action, long-horizon problems?* One example that we have already mentioned above is Gao et al. 2022 above, which seems consistent with our findings. But still, there are valid questions about (a) the computational efficiency of various components of the algorithm and (b) the possibility of a theoretical extension to countable or continuous state spaces. We address (a) to some extent in the paper: computing η in high dimensions is outlined in the Discussion and computing the projection in high dimensions is outlined in Appendix H. Regarding (b), it seems that Theorem 1 holds for countably-infinite state and action spaces (the only change is in the proof of Proposition 1 and Lemma 1). In a continuous setting, the theory becomes more involved, and might require more complex tools - for example, there are some issues regarding the convexity of the space. (Note most of our results just rely on separating the policy from the reward, which the above shows we can still do in the general setting.)
> > The length of the horizon does not have any impact on our theory, but it does increase the computational requirements for computing the occupancy measure (because of the lower rate of convergence). We also investigate the impact of the discount factor on the prevalence of Goodharting in Appendix G, Plot 16c.
> >
> > We hope that our contributions have been made clearer now, and kindly ask the reviewer to consider revising their score if they deem it appropriate.

---

### Meta-Review · Area_Chair_vo1n · 2023-12-06

**Metareview:**

This paper contributes a new analysis of the reward misspecification problem in terms of goodheart's law. All the reviewers found it to be interesting and no major issues emerged in the discussion.

**Justification For Why Not Higher Score:**

None of the reviewers really championed this paper as being super important.

**Justification For Why Not Lower Score:**

None of the reviewers argued there was anything major wrong with it.

---

### Decision · Program_Chairs · 2024-01-16

Accept (poster)